# SHOC2 phosphatase-dependent RAF dimerization mediates resistance to MEK inhibition in RAS-mutant cancers

Greg G. Jones[1], Isabel Boned del Río[1], Sibel Sari[1], Aysen Sekerim[1], Lucy C. Young[1], Nicole Hartig[1], Itziar Areso Zubiaur[1], Mona A. El-Bahrawy[2], Rob E. Hynds [1], Winnie Lei[1], Miriam Molina-Arcas[3], Julian Downward [3,4] & Pablo Rodriguez-Viciana[1]

Targeted inhibition of the ERK-MAPK pathway, upregulated in a majority of human cancers, has been hindered in the clinic by drug resistance and toxicity. The MRAS-SHOC2-PP1 (SHOC2 phosphatase) complex plays a key role in RAF-ERK pathway activation by dephosphorylating a critical inhibitory site on RAF kinases. Here we show that genetic inhibition of SHOC2 suppresses tumorigenic growth in a subset of KRAS-mutant NSCLC cell lines and prominently inhibits tumour development in autochthonous murine KRAS-driven lung cancer models. On the other hand, systemic SHOC2 ablation in adult mice is relatively well tolerated. Furthermore, we show that SHOC2 deletion selectively sensitizes KRAS- and EGFR-mutant NSCLC cells to MEK inhibitors. Mechanistically, SHOC2 deletion prevents MEKi-induced RAF dimerization, leading to more potent and durable ERK pathway suppression that promotes BIM-dependent apoptosis. These results present a rationale for the generation of SHOC2 phosphatase targeted therapies, both as a monotherapy and to widen the therapeutic index of MEK inhibitors.

[1] University College London Cancer Institute, London WC1E 6DD, UK. [2] Department of Histopathology, Imperial College London, Du Cane Road, London W12 0NN, UK. [3] The Oncogene Biology Lab, The Francis Crick Institute, 1 Midland Road, London NW1 1AT, UK. [4] Lung Cancer Group, Division of Molecular Pathology, The Institute of Cancer Research, 237 Fulham Road, London SW3 6JB, UK. Correspondence and requests for materials should be addressed to P.R-V. (email: p.rodriguez-viciana@ucl.ac.uk)

Oncogenic mutations in RAS genes are found in over 30% of human cancers including lung, colon and pancreatic adenocarcinomas. In addition, RAS proteins play a key role in many more cancers through indirect activation, for example, as a result of aberrant signalling by receptor tyrosine kinases (RTKs), or by inactivation of negative regulators such as the NF1 tumour suppressor gene. Non-small cell lung carcinoma (NSCLC), the leading cause of cancer mortality worldwide, has known driver mutations of nodes on this pathway in ~75% of cases, including: ~30% KRAS, ~11% EGFR, ~7% BRAF and ~11% NF1 mutations[1].

RAS proteins have been challenging drug targets and extensive efforts have focused on targeting RAS-effector pathways as a more tractable alternative[2,3]. Multiple lines of evidence including mutual exclusivity of RAS- and BRAF-mutations in many cancers highlight the RAF-MEK-ERK kinase cascade (ERK-MAPK pathway) as a key effector of RAS oncogenic properties and multiple small molecule inhibitors of this pathway have been developed[4]. However, RAS-driven tumours remain intractable to targeted therapies.

RAF and MEK inhibitors have been approved for the treatment of BRAF V600E/K-mutant melanoma but only show transient clinical benefit due to the rapid onset of resistance. Current RAF inhibitors are contraindicated for the treatment of RAS-driven tumours as they promote RAF dimerization and ERK-activation in these cells[5,6].

MEK inhibitors (MEKi) are highly selective due to their allosteric mechanism of action but have shown minimal clinical efficacy against RAS-driven tumours[7,8]. This is mainly due to drug resistance and toxicity. The ERK-pathway is regulated by negative feedbacks at multiple levels including phosphorylation of negative regulatory sites in RTKs, as well as in RAF kinases that inhibit RAF dimerization and binding to RAS[9–12]. Relief of these negative feedbacks by pharmacological ERK-pathway inhibition results in signalling rebound and intrinsic resistance. In addition, the ERK-pathway is a key mediator of G1/S transition and MEKi's have a predominantly cytostatic response that likely facilitates acquisition of drug resistance mechanisms[13]. Strikingly, in both RAS- and BRAF-mutant cells, most resistance mechanisms lead to ERK-pathway reactivation, highlighting the strong 'oncogene addiction' of these cancers to ERK-signalling. However, the potent pathway suppression required for antitumor activity is limited by the dose of MEKi that can be administered safely because of on target toxicity[14,15]. ERK-activity is essential for normal tissue homeostasis, and systemic ablation of MEK1/2 or ERK1/2 genes in adult mice leads to death of the animals from multiple organ failure within 2–3 weeks, even under conditions of partial inactivation[16], highlighting the difficulties of inhibiting the ERK-pathway with a therapeutic index. In order to effectively harness the addiction of RAS-mutant cancers to ERK-signalling into viable therapies, new signalling nodes as well as strategies to improve the therapeutic index of current inhibitors are needed[17–21].

Activation of RAF Kinases is a highly complex process where RAS-GTP binding to the RAS Binding Domain (RBD) of RAF is only the initial step. Dephosphorylation of a conserved inhibitory site in the N-terminal regulatory domain (ARAF S214, BRAF S365, CRAF S259, hereby referred as 'S259') provides an additional key activating input that facilitates 14–3–3 dissociation and RAF dimerization[22–25]. 'S259' RAF dephosphorylation is mediated by a ternary phosphatase complex comprised of SHOC2, MRAS and PP1 (SHOC2 phosphatase complex)[26]. Gain-of-function mutations in the RASopathy Noonan Syndrome in SHOC2, MRAS and PP1, as well as in CRAF (that cluster around S259), underscore the key role of the SHOC2 complex in RAF and ERK-pathway regulation[27–34].

In this study we show that genetic inhibition of SHOC2 suppresses tumour development and elongates overall survival in KRAS-driven Lung adenocarcinoma (LUAD) mouse models, as well as inhibiting tumorigenic growth in a subset of KRAS- and EGFR-mutant human cell lines. Furthermore, by preventing feedback-induced RAF dimerization, combined SHOC2 and MEK inhibition leads to more potent and sustained suppression of ERK-signalling that turns an otherwise reversible cytostatic response to MEKi's into cell death in RAS-mutant cells. Crucially, SHOC2 deletion is well tolerated, both at the cellular level in tissue culture and at the organismal level in adult mice, suggesting a unique potential to provide a therapeutic index. Our study uncovers new insights into SHOC2 biology and reveals inhibition of the SHOC2 phosphatase complex, alone and in combination with MEKi's, as a therapeutic strategy to treat RAS- and EGFR-mutant cancers.

## Results

**SHOC2 inhibition perturbs tumour growth in lung cancer models.** To study the in vivo role of SHOC2 we have developed two mouse models of SHOC2 inactivation using Knock-out (KO) or Knock-in (KI) approaches (Fig. 1a). Because constitutive SHOC2 deletion is embryonically lethal as shown by us and others[35] we used a conditional strategy whereby SHOC2 alleles are inactivated upon cre-mediated recombination. A KO model was generated by flanking exon 4 of SHOC2 with loxP sites, whereas a KI model for the SHOC2-D175N point mutant was generated using a minigene strategy whereby wild-type SHOC2 is expressed in a cDNA configuration under its endogenous promoter and replaced after cre-mediated recombination by the mutant D175N allele, unable to form a complex with MRAS and PP1[26]. To validate our genetic models, MEFs derived from Shoc2$^{fl/fl}$;CreER$^{T2}$, and Shoc2$^{D175N}$;CreER$^{T2}$ compound mice were generated and analysed by western blot (Fig. 1b).

To evaluate the effect of SHOC2 inhibition in vivo in an autochthonous lung cancer model we used the LSL-Kras$^{G12D}$ model, alone (K model) or in combination with a LSL-Trp53$^{R172H}$ allele (KP model) that develops a more severe phenotype compared to the K model[36,37]. In both models, activation of Kras, and inactivation of Shoc2 and p53, were achieved by intranasal delivery of adenovirus-expressing cre.

Shoc2$^{fl/fl}$ and control mice from K and KP models were sacrificed 6-months after adeno-cre delivery and tumour burden quantified by lung weight and H&E histology. SHOC2 inactivation, using either KO or KI models, significantly decreased overall tumour burden, (Fig. 1c–e) and significantly, prolonged overall survival in both K and KP animals (Fig. 1f, g). Importantly, when recombination of the floxed Shoc2 allele was analysed in remaining tumour nodules from Shoc2$^{fl/fl}$ KP mice, a band for the unrecombined Shoc2$^{fl}$ allele could be detected to various levels in a majority of these tumours (Fig. 1h) suggesting that at least a significant proportion of Shoc2$^{fl/fl}$ tumours after 6-months are 'escapers'[16,38,39] further underscoring the key requirement for SHOC2 in lung tumour development.

To study SHOC2 function in adult tissue homeostasis, Shoc2 KO and KI mice were crossed with animals carrying an inducible ubiquitously expressed CreER$^{T2}$ recombinase[40]. To induce SHOC2 deletion systemically in vivo, 8- to 12-week old mice were treated with tamoxifen by oral gavage under conditions where high level (>80%) recombination efficiency was observed 8-weeks after treatment across all tissues examined, except brain (Fig. 1i, j). At this time none of these mice showed signs of toxicities, as determined by normal body weight gain or signs of ill health compared to control animals. Thus whereas SHOC2 is required for KRAS-driven lung tumour formation,

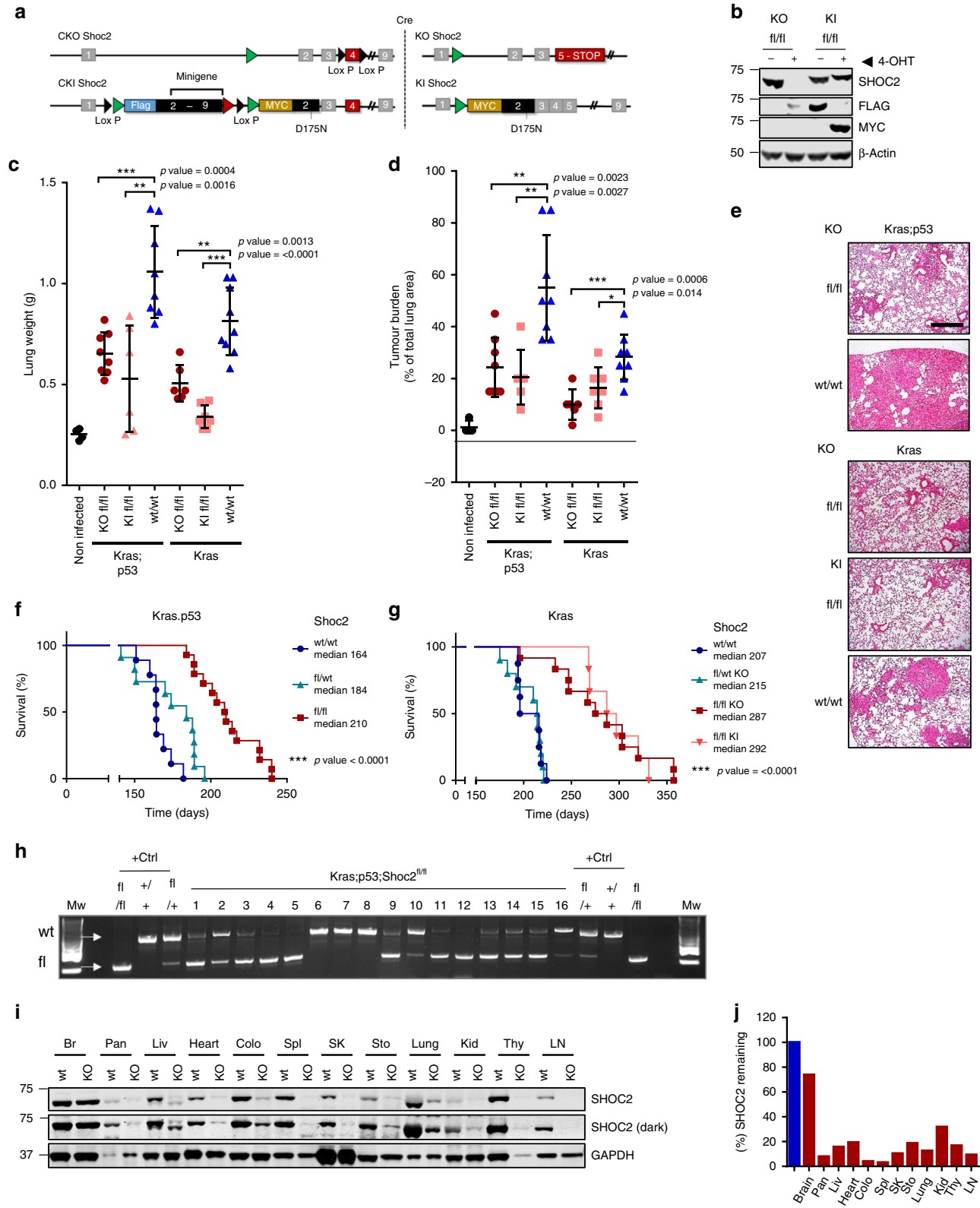

systemic SHOC2 inhibition is well tolerated in adult mice, up to 8–12weeks.

**SHOC2 is required for tumorigenic growth of RAS-mutant cell lines**. To further study the role of SHOC2 in the context of lung tumorigenesis, both shRNA and CRISPR/CAS9 approaches were used to deplete SHOC2 in a panel of human NSCLC cell lines

(Supplementary Table 1). Serum-starved SHOC2 knock-out (KO)/Knock-down (KD) cells consistently had higher basal levels of phospho-S365 BRAF and showed impaired EGF-induced S365/S259 B/CRAF dephosphorylation and MEK and ERK phosphorylation. In contrast, RAS activation or S338 CRAF phosphorylation were unaffected by SHOC2 downregulation (Fig. 2a, Supplementary Fig. 1a, b). Furthermore, re-expression of

**Fig. 1** SHOC2 ablation inhibits tumour growth in both the KRAS$^{G12D}$ and more malignant KRAS$^{G12D}$;TP53$^{R172H}$ Lung adenocarcinoma (LUAD) mouse models. **a** Genetic strategy for the generation of Shoc2$^{fl/fl}$ KO and Shoc2$^{D175N}$ KI mouse models. **b** Validation of recombination strategy. E6-immortalised Shoc2$^{fl/fl}$ CreER$^{T2}$ MEFs were treated with 1 μg/ml 4-OHT for 7-days and lysates analysed by western blot. SHOC2 ablation perturbs growth of Kras$^{G12D}$ and Kras$^{G12D}$;p53$^{R172H}$ mouse LUAD. **c** Lung weight from indicated genotypes 24-weeks post AdenoCre infection. Significance is determined using a two tailed $t$-test * = < 0.05 ** = < 0.01 *** = < 0.001. **d** Lung sections from (**c**) were stained with H&E and quantified for tumour burden as a % of total lung area. Significance as above. **e** Representative H&E images from (**d**). Scale bar = 500 μm. **f** Kaplan-Meier curve of mice with indicated genotypes from the Kras;p53 mouse model. Statistics were determined by log-rank test * = < 0.05 ** = < 0.01 *** = < 0.001. $n$ = 9–14 animals per group **g** As (**f**) with Kras mouse model. $n$ = 6–12 animals per group. **h** Incomplete SHOC2 recombination ('escapers') may account for a significant number of tumours arising in Shoc2$^{fl/fl}$ mice. Cre-mediated recombination was analysed by PCR in the largest nodules isolated from lungs of KRAS$^{G12D}$;p53$^{R172H}$;Shoc2$^{fl/fl}$ mice 6-months post AdenoCre infection. **i** Systemic Shoc2 deletion in adult mice is tolerated. 6- to 8-week-old Shoc2$^{fl/fl}$ CreERT2 mice were subject to treatment of 80 mg/kg of Tamoxifen by oral gavage for 10-days, and tissues harvested 8-weeks later to assess Shoc2 protein levels. Representative figure from $n$ = 5 animals. **j** Quantification of SHOC2 levels relative to GAPDH loading control in (**i**)

SHOC2 wild type (wt), but not SHOC2 D175N, which is defective for phosphatase complex formation with MRAS and PP1[26,33] decreased P-S365 BRAF levels and fully rescued MEK and ERK phosphorylation by EGF (Fig. 2a). This is consistent with a role for SHOC2 in ERK pathway regulation by specifically dephosphorylating the 'S259' inhibitory site in RAF kinases[26].

SHOC2 ablation had no effect on proliferation in 2D-adhered cultures in any KO cell line tested, (Fig. 2b, Supplementary Fig. 2a, f) consistent with SHOC2 not being essential for anchorage-dependent proliferation[41]. In contrast, anchorage-independent proliferation as spheroids was impaired in the absence of SHOC2 in a subset (3 out of 6) of KRAS-mutant cell lines (Fig. 2c, d). SHOC2 ablation also selectively inhibited spheroid growth in isogenic KRAS$^{G12V}$ transformed MEFs (Supplementary Fig. 2b, c). Defective spheroid growth was rescued in KO cells by re-expression of WT-SHOC2, but not SHOC2-D175N (Fig. 2e), consistent with a requirement for SHOC2's role within a 'S259' RAF holophosphatase complex. Defective spheroid growth correlated with tumour formation in subcutaneous xenograft assays, as SHOC2 suppression strongly suppressed xenograft growth of H358 '3D sensitive' cells (Fig. 2f) but did not have a significant effect on A427 '3D resistant' (Fig. 2g) cells. Although SHOC2 deletion had little effect on subcutaneous xenograft growth of A427 cells, it strongly inhibited their implantation and/or growth in lung colonization experiments (Fig. 2h). Taken together with in vivo studies in Fig. 1, this data shows the contribution of SHOC2 to the tumorigenic properties of RAS-mutant cells is dependent on both the cell type and the context.

To understand the preferential requirement for SHOC2 on anchorage-independent growth, we probed lysates of cells grown in adhered or suspension culture conditions for markers of ERK- or PI3K/AKT-signalling. AKT activation was downregulated similarly in both parental and KO cells growing in suspension consistent with previous reports[42,43]. On the other hand, whereas basal ERK-pathway signalling was largely unaffected in SHOC2 KO cells in adhered cultures, it was significantly impaired in suspension cultures (Fig. 2i). Thus, in the context of oncogenic RAS, SHOC2 preferentially contributes to ERK signalling under anchorage-independent conditions.

We noted that cell lines resistant to SHOC2 depletion for spheroid growth (A421, H460, A549) have inactivating mutations in the LKB1/STK11 tumour suppressor gene (Supplementary Table 1) as well as retaining higher AKT phosphorylation levels in suspension (Fig. 2i) suggesting possible molecular mechanisms of resistance to SHOC2 ablation for anchorage-independent growth. Re-expression of LKB1 in A427 LKB1-null cells failed to render them sensitive to SHOC2 depletion, whereas conversely, LKB1 knockdown failed to overcome SHOC2 requirement for spheroid growth in H358 '3D sensitive' cells (Supplementary Fig. 2d, e). Thus, LKB1 status alone does not appear to determine

sensitivity to SHOC2 for 3D growth. On the other hand, ectopic expression of membrane associated, constitutively activate AKT (Myr-AKT1), had no effect on 'SHOC2 3D resistant' A549 cells but fully rescued spheroid growth in 'SHOC2 3D sensitive' H358 and H1792 SHOC2 KO cell lines (Fig. 2j–l, Supplementary Fig. 2f-i). Thus, at least in some contexts, increased AKT signalling may help to overcome SHOC2's requirement for 3D growth in RAS-mutant cells.

**SHOC2 deletion sensitizes KRAS- and EGFR-mutant cells to MEKi's.** In order to broaden SHOC2's properties as a possible therapeutic target in NSCLC, we set out to identify potential synthetic lethal interactions with small molecule inhibitors by screening in the presence or absence of SHOC2. A screen of a candidate panel of small molecule inhibitors revealed a potent and selective sensitization to all MEKi's in the study, upon SHOC2 deletion, in both H358, (Fig. 3a) and A549 (Supplementary Fig. 3a) KO cells. Cells growing in suspension were more sensitive to MEKi than in adhered cells (Supplementary Fig. 3b), in agreement with higher KRAS- and ERK-signalling dependency in suspension culture[44,45]. However, SHOC2 deletion lowered IC50 values to MEKis similarly in both adhered and suspension culture conditions (Supplementary Fig. 3b) and so viability assays were performed under adhered conditions.

We next performed similar experiments in additional NSCLC cell lines encompassing a wider range of driver mutations found in human LUAD, including KRAS, EGFR and BRAF, as well as cells without known driver mutations in the ERK-pathway, (WT). SHOC2 ablation sensitised all KRAS- and EGFR-mutant NSCLC cell lines tested, except one (H727), to the MEK inhibitors Selumetinib (Fig. 3b, c, Supplementary Fig. 4a, c) and Trametinib (Fig. 3c, Supplementary Fig. 3c), but had no effect on the response of BRAF-mutant or WT cell lines. Interestingly the H727 cell line, which is extremely sensitive to MEKi, harbours a potential activating mutation in ARAF (A285D)[46] in addition to KRAS$^{G12V}$.

Sensitization to MEKi's was also seen in RAS-mutant cancer cells derived from other tissue types such as MDA-MB-231 (TNBC, KRAS$^{G13D}$), HCT116 (Colorectal, KRAS$^{G13D}$), Pa-Tu8092 (Pancreatic, KRAS$^{G12V}$) and SK-MEL-2 (Melanoma, NRAS$^{Q61L}$) but not BRAF$^{V600E}$ HT-29 colorectal cells (Fig. 3b, Supplementary Fig. 4c). No significant changes were seen with any other inhibitors tested, including those targeting RAF and ERK-nodes of the ERK-pathway, in both viability and colony formation assays (Fig. 3c, d). Significantly, results observed in RAS-mutant versus WT human cancer cell lines, were recapitulated upon SHOC2 inactivation in isogenic NL20 immortalized, nontumorigenic human bronchial epithelial cells (Fig. 3e, f), as well as in MEFs derived from Shoc2$^{fl/fl}$;CreER$^{T2}$ mice (Fig. 3g), where SHOC2 ablation selectively sensitized cells to the MEKi

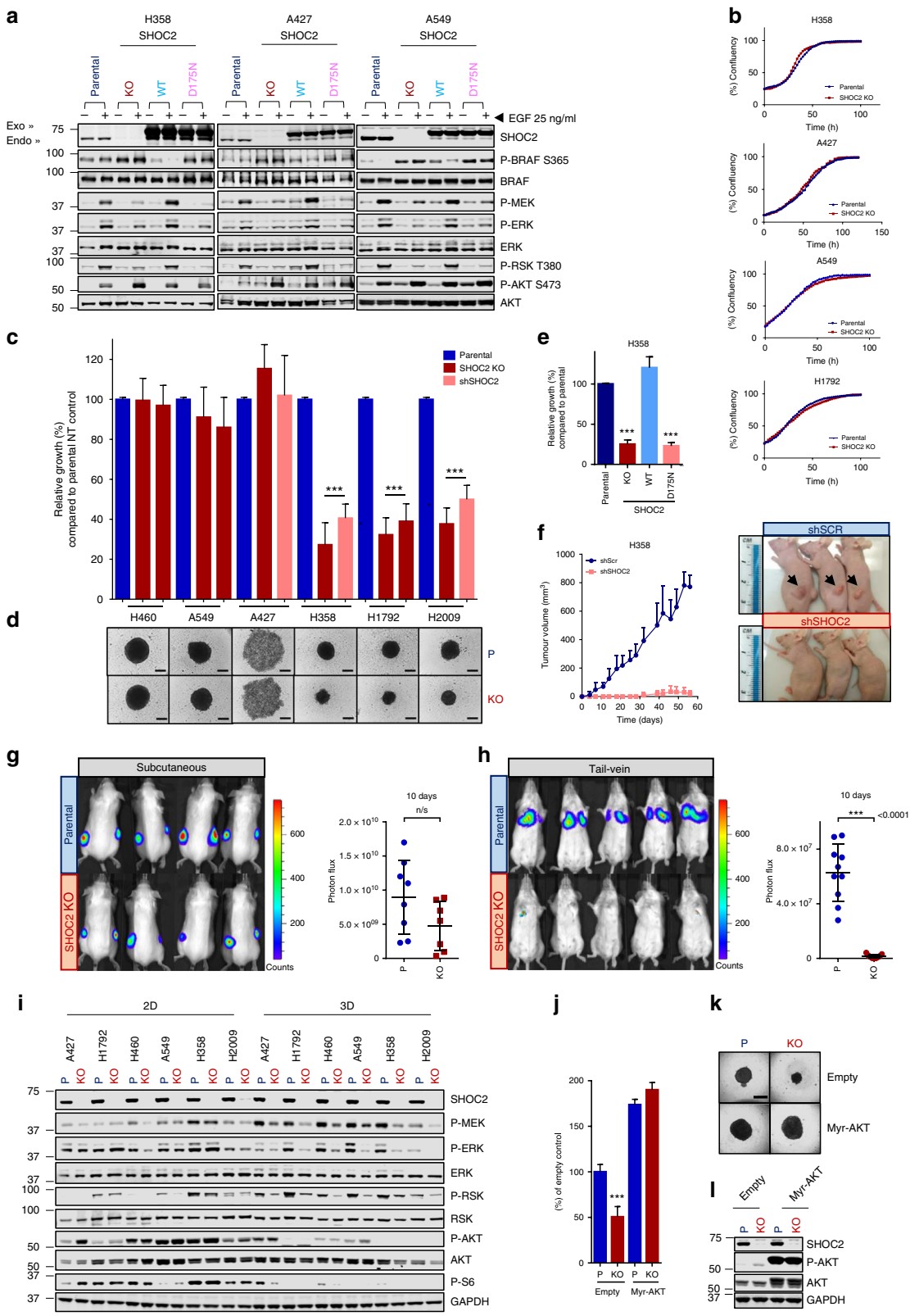

Selumetinib (but not upon PanRAFi LY3009120) only upon ectopic expression of KRAS$^{G12V}$.

Sensitization to MEKi's upon SHOC2 ablation was fully rescued by re-expression of WT-SHOC2, but not SHOC2-D175N (Fig. 3h, Supplementary Fig. 3d). Furthermore, overexpression of SHOC2-independent, phosphorylation-deficient

'S259A' RAF-mutants, also diminished the enhanced MEKi sensitivity (Fig. 3i, j, Supplementary Fig. 3e-f). Taken together, these observations suggest genetic inhibition of SHOC2 potently and selectively sensitizes RAS- and EGFR-mutant cells from different tissue types to MEKis, in a manner consistent with SHOC2's function as a RAF 'S259' phosphatase.

**Fig. 2** SHOC2 is required for the tumorigenic growth of RAS-mutant cell lines. **a** WT- but not D175N-SHOC2 is sufficient to rescue impaired ERK-pathway response to EGF in SHOC2 KO cells. H358/ A427/A549 parental and SHOC2 KO cells stably expressing WT or D175N-SHOC2 were treated with 25 ng/ ml EGF for 10 min and lysates probed as indicated. **b** SHOC2 deletion has no effect on adhered cell growth of RAS-mutant NSCLC cell lines. Incucyte growth curves. Representative of $n = 3$. **c** SHOC2 KO/KD perturbs anchorage-independent growth of a subset of RAS-mutant NSCLC cell lines. Parental/ SHOC2 KO clones or shSHOC2 transduced cells were seeded under anchorage-independent conditions and growth determined at Day 5 by alamar blue staining (mean ± SD) ($n = 4$). ∗$p < 0.05$, ∗∗$p < 0.01$ or ∗∗∗$p < 0.001$ as determined by two tailed $t$-test. **d** P/C images of representative spheroids measured in (**c**). Scale bar = 200 μm. **e** Inhibition of anchorage-independent growth in SHOC2 KO H358 cells is rescued by re-expression of WT-, but not D175N-SHOC2. Cells described in (**a**) were seeded as (**c**). **f** SHOC2 depletion prevents xenograft growth of H358 cells. 5*10$^6$ shSCR or shSHOC2 cells were injected subcutaneously per flank in athymic nude mice ($n = 5$ animals per group). **g**–**h** SHOC2 is dispensable for subcutaneous tumour growth of A427 cells, but is required for growth in orthotopic lung colonization assays. 2.5*10$^6$ Parental or SHOC2 KO cells stably expressing luciferase were injected into SCID mice (**g**) subcutaneously, $n = 4$ animals per group, or (**h**) into lateral tail vein, $n = 10$ animals per group. Tumour burden was assessed after 10 days by bioluminescence imaging (mean ± SD). **i** SHOC2 selectively contributes to ERK signalling under anchorage-independent conditions. Parental and SHOC2 KO cells were seeded in regular or poly-HEMA coated plates for 24 h and lysates probed as indicated. **j** Inhibition of anchorage independent growth on SHOC2 depletion in RAS-mutant cells is rescued by MYR-AKT expression. Parental or SHOC2 KO H358 cells with stable expression of Empty vector or MYR-AKT were seeded in low-attachment plates and growth determined at Day 5 by alamar blue (mean ± SD, $n = 4$) **k** P/C images of representative spheroids measured in (**j**) at D5. Scale bar = 200 μm. **l** Lysates of cells in (**j**) were probed with indicated antibodies

**SHOC2 is required for ERK activation induced by MEKi's.** To characterize the molecular mechanisms of sensitization to MEKis by genetic inhibition of SHOC2, we measured ERK pathway activation using a dose response with the MEKi Selumetinib in H358 cells, either acutely (0.5 h), or for a longer period (12 h) (Supplementary Fig. 5). IC50 values for MEK and ERK phosphorylation were unchanged after acute treatment with Selumetinib (Supplementary Fig. 5a-c). On the other hand, following 12 h of MEKi treatment (Supplementary Fig. 5d–f) there is a dose-dependent increase in MEK phosphorylation consistent with signalling rebound upon feedback relief[47,48]. Notably this dose-dependent rebound in P-MEK was impaired in SHOC2 KO cells and this correlated with more durable suppression of ERK phosphorylation at lower MEKi doses (Supplementary Fig. 5d–f).

To study the role of SHOC2 in feedback relief-induced ERK pathway reactivation a time course of Selumetinib (Fig. 4a) and Trametinib (Supplementary Fig. 6a) was performed over 72 h in both KRAS- (A549, H358) and EGFR-mutant (HCC4006) parental and SHOC2 KO cells. MEKi treatment promotes dephosphorylation of S365 BRAF peaking after ~12 h of treatment (Fig. 4a, b). MEK and ERK rebound phosphorylation were readily detected after 12 h of treatment (Fig. 4a, c) and increased over time in a cell line dependent-manner. Strikingly, in SHOC2 KO cells, MEK rebound phosphorylation was strongly impaired and ERK pathway activity remained potently suppressed over the 72 h treatment period which correlated with accumulation of BIM protein compared to parental cells.

We next performed 'wash-out' experiments to analyse pathway reactivation after acute MEKi withdrawal (Fig. 4d–h). After Selumetinib wash-out in KRAS mutant A549 or A427 cells, ERK phosphorylation was detected in control cells by 10 min, peaked at 30 min and approached basal levels by 180 min. This response is consistent with the phosphorylated (but inactive when inhibitor-bound) MEK, leading to a wave of ERK phosphorylation upon MEKi removal before new feedbacks regenerate a steady state. ERK pathway reactivation at the level of P-MEK, P-ERK and P-RSK is strongly impaired in SHOC2 KO cells but high basal RAS-GTP or P-S338 CRAF levels were unaffected (Fig. 4d). P-MEK rebound and delayed ERK reactivation on MEKi treatment is fully rescued by re-expression of WT, but not D175N SHOC2 (Supplementary Fig. 6b-c). In contrast to MEKi, MEK and ERK-phosphorylation, as well as BIM protein levels after treatment with the PanRAFi LY3009120 are independent of SHOC2 (Fig. 4f) which correlates with the sensitization of SHOC2 KO/KD cells to MEK but not to PanRAFi's (Fig. 3).

Similar effects are seen across multiple KRAS- and EGFR-mutant cell lines with both Selumetinib and Trametinib (Fig. 4e, Supplementary Fig. 6a-e). On the other hand, P-MEK rebound and ERK reactivation are only minimally affected in WT H520 and H522 cells in the absence of SHOC2 (Fig. 4g, h). Furthermore, in wash-out experiments in Shoc2$^{fl/fl}$;CreER$^{T2}$ MEFs, ectopic expression of KRAS$^{G12V}$ induces a preferential dependency on SHOC2 for P-MEK rebound and ERK pathway reactivation after MEKi withdrawal (Supplementary Fig. 6f-h). Taken together these results strongly suggest that KRAS- and EGFR-mutant cells are selectively dependent on SHOC2 for feedback-mediated pathway reactivation upon MEK inhibition.

**SHOC2 is required for RAF dimerization induced by MEKi's.** MEKi treatment stimulates BRAF-CRAF dimerization in KRAS-mutant cells[49]. Therefore, to explore molecular mechanisms of the SHOC2-dependent feedback reactivation response, we performed wash-out experiments as described above and analysed RAF dimerization by co-immunoprecipitation. Treatment with both Selumetinib and Trametinib led to a strong induction of BRAF dimerization with both CRAF and ARAF in control cells that was almost completely inhibited in SHOC2 KD/KO RAS- and EGFR-mutant cell lines (Fig. 5a, b). This correlates with (i) decreased MEKi-induced P-MEK accumulation in the absence of SHOC2, (ii) dampened rebound phosphorylation of ERK and ERK substrates, and (iii) more durable pathway inhibition after MEKi withdrawal (see model Fig. 5h). In contrast however, the potent RAF dimerization induced by the RAF inhibitor LY3009120[50,51] is not affected by loss of SHOC2 (Fig. 5c), in agreement with LY3009120-induced ERK-reactivation being independent of SHOC2 (Fig. 4f). Intriguingly, the ERK inhibitor LY3214996 induced potent BRAF-CRAF dimerization and MEK rebound phosphorylation that was SHOC2 dependent as seen with MEKi (Fig. 5d). However, unlike MEKi (but like RAFi) ERK phosphorylation was unaffected by SHOC2 loss. Thus sustained inhibition of ERK phosphorylation upon SHOC2 loss is selectively observed when the RAF-MEK-ERK cascade is inhibited at the level of MEK, but not RAF or ERK, which in turn correlates with sensitization in viability assays (Fig. 3).

To further assess the role of RAF dimerization, wash-out experiments were performed in cells where expression of individual RAF isoforms was inhibited with siRNAs (Fig. 5e). Knockdown of BRAF or CRAF, but not ARAF partially perturbed both P-MEK rebound upon Selumetinib treatment and ERK and RSK reactivation after Selumetinib withdrawal (Fig. 5e, f)[52]. Effects of siRNAs on signalling rebound correlate well with

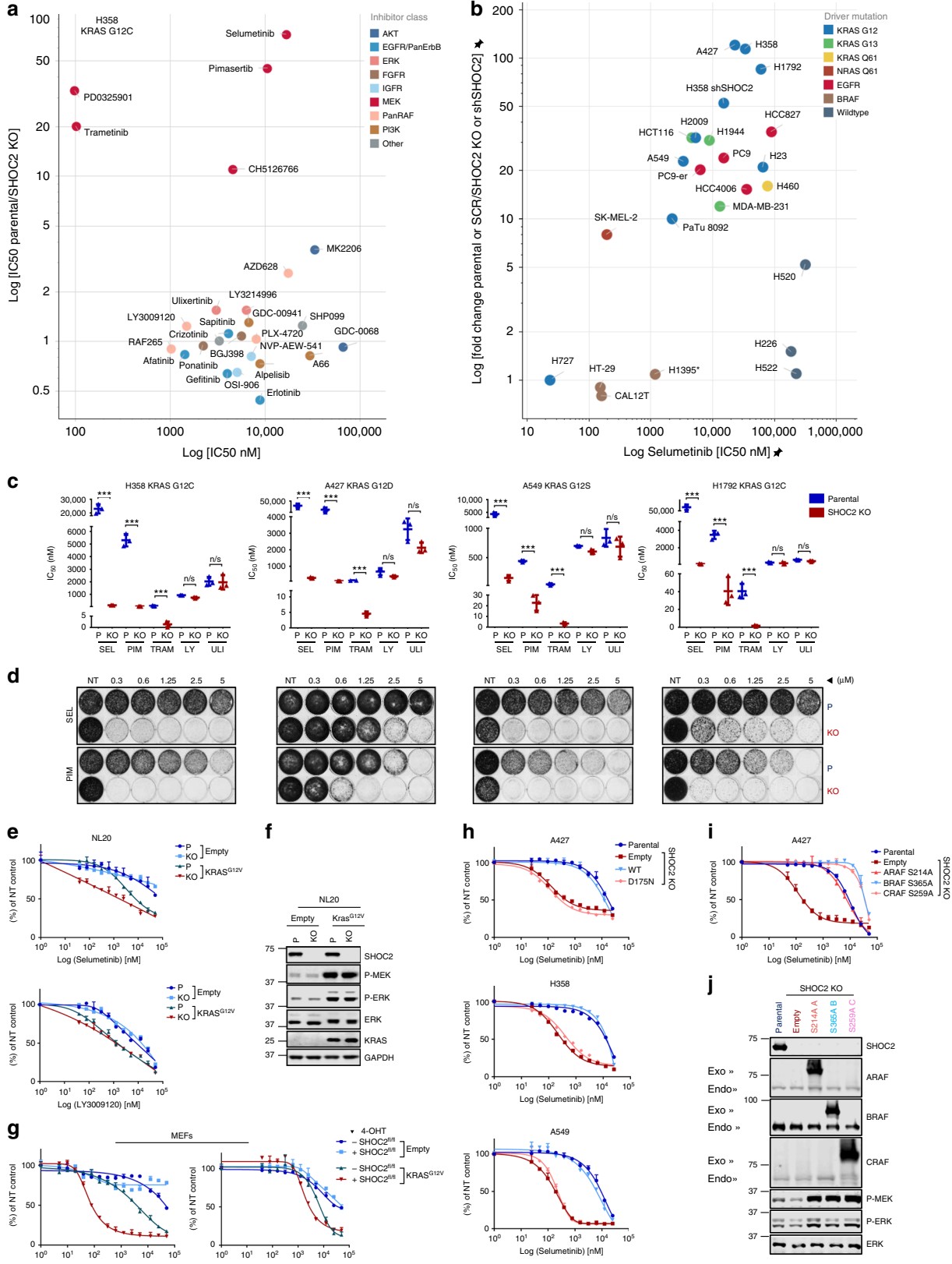

sensitization in viability assays (Fig. 5g), as knockdown of either BRAF or CRAF, but not ARAF, sensitise KRAS-mutant NSCLC cells to MEKi although again not as strongly as SHOC2 knockdown. Collectively, this data highlights the requirement of SHOC2-dependent BRAF-CRAF dimerization in mediating signalling rebound upon MEKi treatment.

**Combined SHOC2 and MEK inhibition promotes apoptosis in RAS-mutant cells.** To characterize the nature of the increased sensitivity to MEKi's in the absence of SHOC2, and assess whether cytostatic vs cytotoxic effects were at play, growth curves were generated from cells cultured in the presence of MEKi for 96 h, after which the MEKi was replaced with fresh media.

**Fig. 3** SHOC2 deletion sensitizes KRAS- and EGFR-mutant cells to MEKi's. **a** Viability assays were performed for Parental and SHOC2 KO H358 cells with indicated inhibitors. The resulting IC50 values are plotted (x-axis), and compared against the fold change between the IC50 values determined for the Parental versus SHOC2 KO cells (y-axis). **b** SHOC2 deletion/depletion sensitises RAS- and EGFR-mutant, but not BRAF-mutant or wild-type cell lines to the MEKi Selumetinib. IC50 values plotted as (**a**) for either Parental versus SHOC2 KO cells or shSCR versus shSHOC2 cells. *Due to incomplete knockdown with shRNA, SHOC2 was inhibited in H1395 by siRNA. *er-Erlotinib resistant. Cell lines are grouped by colour code based on driver mutation. **c** SHOC2 deletion selectively sensitises KRAS-mutant NSCLC lines to MEK, but not PanRAF or ERK inhibitors. Replicate IC50 values are plotted for the indicated cell lines comparing Parental versus SHOC2 KO cells (mean ± SD) (n = 3). Significance is determined using a two tailed t-test ∗p < 0.05, ∗∗p < 0.01 or ∗∗∗p < 0.001. **d** SHOC2 deletion sensitises RAS-mutant cells to MEKi's in colony formation assays. **e** Nontumorigenic human bronchial NL20 epithelial cells were infected with retrovirus expressing KRAS[G12V] or an empty vector control and viability assays performed with the MEKi Selumetinib or the PanRAFi LY3009120. **f** Lysates of Cells from (**e**) were probed with indicated antibodies. **g** E6-immortalized MEFs derived from Shoc2[fl/fl];CreER[T2] mice were infected with retrovirus expressing KRAS[G12V] or an empty vector control and viability assays performed as (**e**). **h** Sensitisation of SHOC2 KO NSCLC cell lines to MEKi's is rescued by re-expression of WT- but not D175N-SHOC2. Viability assays were performed for SHOC2 KO cells after stable expression of WT-SHOC2, SHOC2 D175N or empty vector control. **i** Sensitisation of SHOC2 knockout NSCLC cell lines to MEKi's is rescued by expression of RAF 'S259' phosphorylation-deficient mutants. Viability assays were performed for SHOC2 KO cells after stable expression of S214A ARAF, S365A BRAF, S259A CRAF or empty vector control. **j** Lysates of cells from (**i**) were probed with indicated antibodies

MEKi's inhibit proliferation in a dose-dependent manner with cytotoxic effects inferred only at the highest concentrations from the absence of surviving cells that resumed growth after inhibitor removal. In the absence of SHOC2 however, complete 'cytotoxic' effects were seen at concentrations that only had partial and/or reversible cytostatic effects on control cells (Fig. 6a, b). When apoptosis was measured after 48 h by Annexin V staining, significant cell death was similarly seen in control cells only at the highest concentration of MEKi used. However, in the absence of SHOC2, MEKi-induced apoptosis was more potent and achieved at lower concentrations (Fig. 6c, d and Supplementary Fig. 7a-c).

ERK signalling regulates multiple proteins involved in apoptosis including the pro-apoptotic BH3 proteins BAD and BIM[53,54]. More potent ERK suppression by MEKi in the absence of SHOC2 correlates with increased levels of BIM and cleaved PARP (Fig. 6e, f). BIM has repeatedly emerged as a key mediator of apoptosis induced by targeted inhibitors, including MEKi, in many cancer cells[53,54]. To address the role of BIM in mediating SHOC2 sensitization to MEKi, siRNAs were used to inhibit BIM expression in viability assays. BIM knockdown diminished the sensitization to MEKi seen in SHOC2 KO cells (Fig. 6g and Supplementary Fig. 7e) and completely abrogated MEKi induced apoptosis of SHOC2 KO RAS-mutant cells (Supplementary Fig. 7d), underscoring BIM as a key mediator of increased cytotoxicity upon combined SHOC2 deletion and MEK inhibition.

Increased apoptosis in in vitro assays correlated with marked regressions in an A427 xenograft model. Treatment with a low dose of the MEKi Trametinib (0.4 mg/kg) that only had a cytostatic response in tumours from control cells, caused marked tumour regressions of SHOC2 KD cells that persisted beyond the treatment window (Fig. 6h, i). Taken together, these observations show that concomitant genetic SHOC2 inhibition potentiates the cytotoxic properties of MEKi's in a BIM-dependent manner and increases antitumor efficacy in lung cancer cell lines.

## Discussion

This study highlights a critical role for SHOC2 as part of the MRAS-SHOC2-PP1 phosphatase complex for oncogenic ERK signalling in NSCLC, the leading cause of cancer-related mortality. SHOC2 ablation in both the LSL-Kras[G12D], and more aggressive Kras[G12D]Trp53[R172H] LUAD mouse model, potently suppresses tumour development and extends lifespan, as has been shown previously for B & CRAF ablation in the LSL-Kras[G12D] model[39]. Similar observations were seen in a SHOC2[D175N] KI model, which will more closely phenocopy pharmacological inhibition of the SHOC2 phosphatase complex in the clinic (Fig. 1). On the other hand, SHOC2 ablation appears to be tolerated remarkably well, both in human cells in culture, and crucially, at the organismal level in adult mice, where animals appear normal 8-weeks after efficient systemic genetic ablation (Fig. 1). This is in clear contrast to MEK and ERK core nodes of the pathway where systemic inhibition leads to fatal toxicities in adult mice[16]. SHOC2/Sur-8 was originally identified in C. elegans as a positive modulator of the RTK-RAS-ERK-pathway that unlike RAF/Lin-45, MEK or ERK/Sur-1 genes, is not essential for organ development but potently suppresses the phenotype of mutant RAS or high FGFR signalling[55,56]. Thus, both C.elegans and mouse genetics highlight how, in the context of oncogenic RAS, targeting the SHOC2 regulatory node of the ERK pathway, may have milder toxicity and thus provide better therapeutic margins than targeting core components such as RAF, MEK or ERK.

In human cell lines, SHOC2 is dispensable for anchorage-dependent proliferation, but is required for anchorage-independent spheroid growth and/or tumorigenic properties in KRAS-mutant NSCLC cell lines (Fig. 2). Anchorage-independent growth reveals a SHOC2-dependent contribution to ERK signalling, not observed in basal adhered culture conditions. This suggests there must be redundant and/or SHOC2-independent mechanisms of ERK activation in adhered growth conditions. Integrin signalling is known to provide a crucial contribution to PI3K/AKT pathway activation in adhered culture that is lost in suspension[42,43,57,58], and it is likely that SHOC2-independent mechanisms of ERK activation linked to integrin signalling are similarly lost in suspension. Furthermore, impaired PI3K/AKT activation of RAS-mutant cells cultured in suspension may help unmask SHOC2's contribution to tumorigenic properties in RAS-mutant cells: reduced cooperation from other signalling pathways enhances the dependency on SHOC2-dependent ERK-signalling for anchorage-independent growth (i.e. 'RAS oncogene addiction to SHOC2 in 3D'). Conversely, our data suggests that aberrant signalling by the PI3K/AKT (and/or other) pathway(s) can compensate for loss of SHOC2-dependent ERK-signalling under anchorage-independent conditions, to promote tumorigenic growth in a cell and context-dependent manner (Fig. 2, Supplementary Fig.2). Regardless, SHOC2's contribution to tumorigenic properties in some RAS-mutant human cells lines, as well as to tumor development in a KRAS-driven mouse LUAD model suggests targeting SHOC2 in the clinic may have activity as monotherapy against a subset of RAS-mutant cancers. Genome wide synthetic lethal studies have also shown a preferential dependency of RAS-mutant cells for SHOC2 function[59,60].

Additionally, we show that SHOC2 deletion sensitizes KRAS- and EGFR-mutant NSCLC cell lines specifically to MEK inhibitors. Notably we observe a similar sensitization to MEKi in the

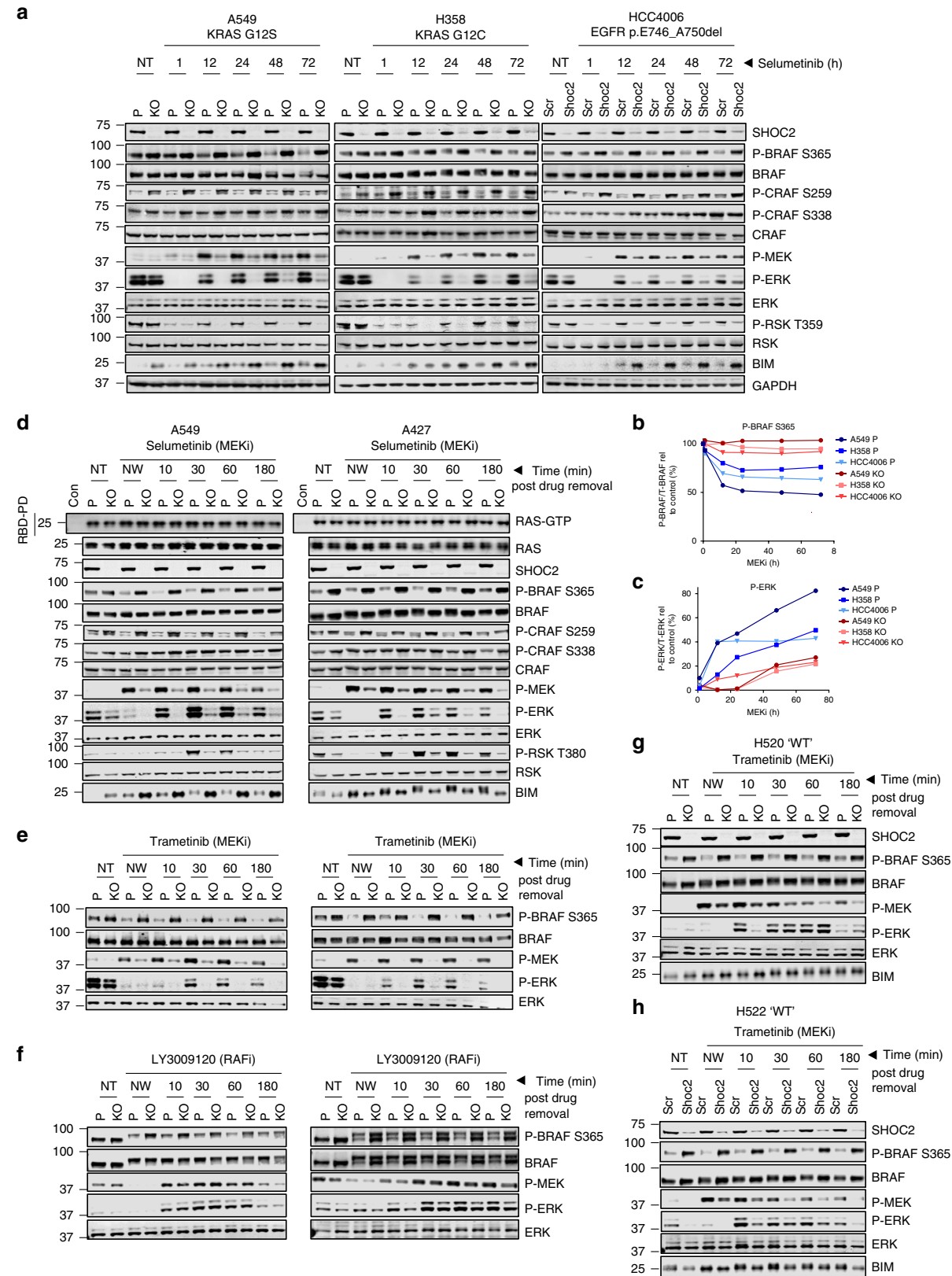

context of oncogenic RAS in isogenic non-transformed bronchial epithelial NL20 cells as well as MEFs (Fig. 3). These observations suggest that rewiring of cellular signalling by oncogenic RAS (or high RAS-GTP levels by RTK signalling) creates a new synthetic lethal interaction for combined MEK and SHOC2 inhibition that could be used as a therapeutic strategy against cancers

with high RAS activity. Mechanistically, our results demonstrate this is due to the requirement for SHOC2 holophosphatase function for RAF dimerization driven by MEKi-induced feedback relief in the context of high basal RAS-GTP levels (Figs 4, 5). This is consistent with a model whereby coordinate inputs provided by (i) direct RAF binding to RAS-GTP and (ii) SHOC2 complex

**Fig. 4** SHOC2 is required for feedback relief ERK activation induced by MEKi's. **a** SHOC2 deletion impairs ERK-reactivation after treatment with Selumetinib. Indicated cells were treated with 1 μM Selumetinib and lysates collected at indicated time points. **b** Quantification of P-BRAF/BRAF over time for cell lines shown in (**a**) relative to NT control. **c** Quantification of P-ERK/ERK over time for cell lines shown in (**a**) relative to NT control. **d–f** SHOC2 deletion impairs MEK, but not PanRAF induced ERK-reactivation. A549 and A427 cells were pre-treated for 12 h with either 1 μM Selumetinib (**d**) / 100 nM Trametinib (**e**) / or 2.5 μM LY3009120 (**f**). Cells were either lysed at this point (NT - Non Treated, NW - Non washed) or the inhibitor was washed-out for the indicated time points before lysate collection. Lysates were used to perform RAS-RBD pull downs and the additional cell lysate probed with indicated antibodies. **g** H520 cells or **h** H522 cells, which have no known driver mutations in the ERK pathway show a reduced dependency on SHOC2 for MEKi-induced ERK-reactivation. Parental or SHOC2 KO H520/H522 cells were treated as (**e**)

mediated 'S259' RAF dephosphorylation is required for RAF dimerization and efficient ERK pathway activation[25,26] (Fig. 5h).

Impaired RAF dimerization in response to MEKi treatment upon SHOC2 deletion correlates with impaired MEK rebound phosphorylation and a deeper and more durable suppression of ERK-signalling after inhibitor withdrawal (Fig. 4a, Supplementary Fig.6a). We have complemented 'inhibitor time courses' with 'inhibitor wash-out' experiments as an experimental paradigm to study ERK reactivation and show that the type of response in both assays correlate well with sensitization to inhibitors in viability assays: In the absence of SHOC2, feedback relief mediated ERK-activation is selectively impaired in KRAS- and EGFR-mutant NSCLC cell lines treated with MEK, but not RAF or ERK inhibitors (Fig. 4).

CRAF is required for ERK-feedback reactivation[52,61]. Here we extend this observation to show that both BRAF and CRAF, but not ARAF knockdown, impair ERK-pathway reactivation and sensitize KRAS-mutant NSCLC cell lines to MEKi, although not as strongly as SHOC2 KD (Fig. 5). A more potent response of SHOC2 depletion compared to single depletion of BRAF or CRAF is consistent with SHOC2 functioning as a PanRAF 'S259' phosphatase. Our data is consistent with a key role for BRAF-CRAF dimers as primary mediators of signalling rebound and resistance to MEKi's. However, a role for ARAF cannot be fully excluded as SHOC2 deletion also prevents ARAF-BRAF dimers. In summary, depending on the node as well as the mechanism of action of the inhibitor, impaired feedback-mediated RAF dimerization and ERK-reactivation in the absence of SHOC2 correlates with increased sensitization in viability assays of RAS- and EGFR-mutant cells and thus provides a good biomarker for SHOC2 sensitization.

The ERK pathway is a key regulator of G1/S transition and MEKi's predominantly exert cytostatic effects, which likely contributes to their poor clinical efficacy and facilitates the selective pressure to acquire resistance mechanisms[53,62]. Importantly, SHOC2 inactivation greatly potentiates apoptosis induced by MEKi in KRAS-mutant NSCLC cells and this correlates with complete cytotoxic responses in tissue culture and with marked tumour regressions in a xenograft model, at MEKi concentrations that otherwise only induce a reversible cytostatic response (Fig. 6).

ERK signalling regulates multiple proteins involved in apoptosis and can control the balance of pro- and antiapoptotic BCL2 proteins to modulate the apoptotic threshold[53,54]. ERK phosphorylation of the pro-apoptotic BH3 proteins BAD and BIM leads to sequestration by 14–3–3 proteins and protein degradation[63–66]. Combined genetic inhibition of SHOC2 and MEK inhibitor treatment cooperate to increase BIM protein levels suggesting a biochemical mechanism to reach the apoptotic threshold (Fig. 4). Furthermore, suppressing BIM expression strongly inhibits sensitization to MEKis upon SHOC2 deletion (Fig. 6). Taken together our data is consistent with a model where the more potent and sustained ERK suppression achieved by co-targeting SHOC2 and MEK allows pro-apoptotic BH3 proteins to accumulate to levels required to induce apoptosis.

As seen with SHOC2, intrapathway dual inhibition (vertical inhibition) at the level of RAF or ERK (MEKi plus RAFi or MEKi plus ERKi) also impairs feedback reactivation, leads to more potent and sustained ERK suppression, promotes tumour regression in preclinical models in RAS-mutant cells[49,61,67]. However, on-target toxicity of pharmacological ERK-pathway inhibition remains the more challenging hurdle for clinical efficacy. It is hard to rationalize how more potent and sustained pathway suppression by vertical inhibition with MEKi + RAFi or MEKi + ERKi combinations can be less toxic than MEKi alone and thus significantly improve therapeutic margins in RAS-driven tumours. In clear contrast to the RAF-MEK-ERK core nodes of the pathway, SHOC2 deletion appears to be tolerated relatively well, both in tissue culture and crucially at the organismal level in mice. Whereas systemic ablation of RAF, MEK or ERK genes in adult mice is not tolerated[16,68] SHOC2 KO mice appear normal 8-weeks after systemic deletion using a similar model of conditional inactivation. Although the toxicity of combined systemic SHOC2 and MEK inhibition in vivo remains to be addressed, our study suggests that uniquely among other pathway nodes for vertical inhibition, co-targeting the SHOC2 holophosphatase may overcome MEKi resistance in RAS-mutant cells and deliver more potent and durable ERK pathway suppression that drives cytotoxic responses at lower MEKi doses with less toxicity.

BRAF-mutant cancer cells are not sensitized to MEKi in the absence of SHOC2, which is consistent with signalling by oncogenic BRAF being independent of RAF dimerization[47,48,69] and therefore of SHOC2 phosphatase function. However, resistance mechanisms to current clinical RAF inhibitors frequently depend on RAF dimerization that are expected to be sensitive to SHOC2 inhibition. Several reports support this possibility[67,70] and thus, SHOC2 could also provide a useful target for combination therapies against BRAF-mutant cancers in some contexts.

Our results highlight SHOC2 as an attractive therapeutic target and provide a rationale for the development of pharmacological inhibitors of the SHOC2 phosphatase complex. Phosphatase inhibitors continue to lag behind kinase inhibitors in drug discovery, but there is emerging evidence that PP1 holophosphatases represent underexplored targets of pharmacological inhibition[71,72]. PP1 functions in complex with over 200 distinct regulatory proteins, with each providing unique properties and substrate specificity to the resulting holophosphatase[73]. Whereas catalytic inhibitors of PP1 will inhibit hundreds of holophosphatases and are toxic to cells, inhibition of specific holophosphatase complexes is expected to inhibit only dephosphorylation of their cognate substrates. There is now indeed proof-of-concept for allosteric inhibition of a PP1 holophosphatase by small molecules targeting its regulatory subunit[74,75]. Inhibitors of the SHOC2 holophosphatase may have activity as single agent in RAS-driven cancers and widen the therapeutic index of MEKi alone or in combination with other targeted agents against the many cancers with high RAS activity.

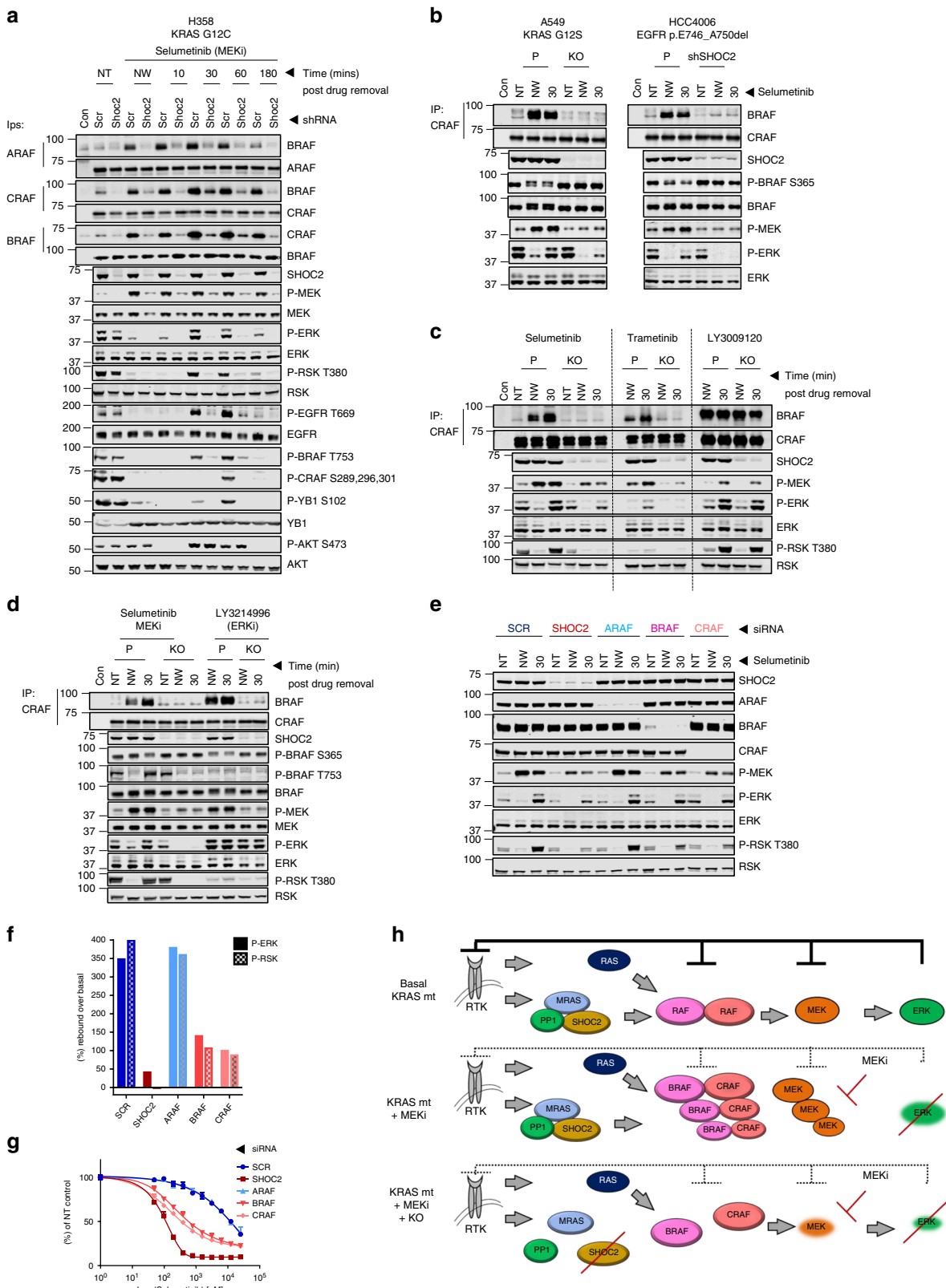

## Methods

**Cell culture and generation of stable cell lines.** HEK293 (UCL) and NSCLC cells obtained from the CRUK Central Cell Services facility (Francis Crick) were cultured in DMEM supplemented with 10% FBS at 37 °C under 5% $CO_2$. NL20 cells, a gift from Charles Swanton's research group were cultured in Ham's F12 medium supplemented with 1.5 g/L sodium bicarbonate, 2.7 g/L glucose, 2.0 mM L-glutamine, 0.1 mM nonessential amino acids, 0.005 mg/ml insulin, 10 ng/ml epidermal growth factor, 0.001 mg/ml transferrin, 500 ng/ml hydrocortisone and 4% fetal bovine serum. For EGF stimulation experiments, cells were serum-starved in DMEM/0% FBS o/n followed by acute treatment with 25 ng/ml EGF.

Lentiviruses shSHOC2 or shSCR were generated by transient transfection of HEK293 cells with the lentiviral construct, pMD.G (VSV-G expresser) and p8.91 (gag-pol expresser) packaging vectors. Cells were transfected 4 h after seeding with plasmid DNA and 1 mg/ml polyethylenimine (PEI, Polysciences) mixed at a 1:4

**Fig. 5** SHOC2 is required for RAF dimerization induced by MEKi's. **a** SHOC2 depletion abrogates MEKi-induced RAF dimerization and impairs ERK pathway reactivation after MEKi withdrawal. shSCR of shSHOC2 H358 cells were pre-treated with 1 μM Selumetinib for 12 h, before the inhibitor was washed-out at indicated time points and lysates used to perform endogenous RAF IPs. (NT - Non Treated, NW - Non washed). Con = IgG control IP. **b** As (**a**) using A549 and HCC4006 cells with a single wash-out time point of 30 min. **c** SHOC2 is required for MEK but not PanRAFi-induced RAF dimerization. Parental and SHOC2 KO H358 cells were pre-treated with 1 μM Selumetinib, 100 nM Trametinib 2.5 μM LY3009120 and subject to endogenous RAF IPs as (**a**). **d** SHOC2 is required for ERK inhibitor induced RAF dimerization. As (**c**), H358 cells were treated with 1 μM Selumetinib and 2 μM LY3214996. **e** B & C but not ARAF knockdown partially diminish MEKi induced signalling rebound and ERK reactivation. H358 cells transfected with indicated siRNAs were treated 3 days later with 1 μM Selumetinib for 12 h before the inhibitor was washed-out for 30 min. (NT - Non Treated, NW - Non washed). **f** Quantification of P-ERK and P-T380 RSK in (**e**). **g** B & C, but not ARAF knockdown partially sensitise H358 cells to Selumetinib. Viability curves for Selumetinib of H358 cells transfected with siRNAs as in (**e**). **h** Schematic to illustrate the requirement of the SHOC2 phosphatase complex for feedback relief ERK-activation on MEKi treatment. ERK activity in RAS-mutant cells is maintained at steady state by negative feedbacks at multiple levels including RTK and RAF pathway nodes. MEKi treatment leads to feedback relief ERK-pathway activation that is both dependent upon RAS-GTP and SHOC2 phosphatase-dependent 'S259' dephosphorylation for RAF dimerization. Following inhibitor withdrawal, release of this 'primed' P-MEK (phosphorylated but unable to activate ERK when inhibitor-bound) generates a wave of ERK phosphorylation that is dampened by negative feedbacks. Even in the presence of mutant RAS in SHOC2 KO cells MEKi induced feedback relief RAF dimerization is prevented, leading to reduced P-MEK rebound and more potent and durable ERK inhibition

ratio in OptiMEM (Life Technologies). Virus-containing medium was harvested 24, 48 and 72 h after transfection and supplemented with 5 μg/ml Polybrene (hexadimethrine bromide, Millipore Sigma). Cells were transduced with lentivirus and where required, selection was carried out with 2.5 μg/ml puromycin (Sigma). Ecotropic retroviruses were generated by transient transfection of the Phoenix ecotropic cell line and virus was collected as above. Cell lines expressing the ecotropic receptor EcoR were generated by transduction with amphotropic EcoR retroviruses and selection with Blasticidin.

SHOC2 knockout (KO) cells were generated by transient transfection with the pSpCas9(BB)-2A-GFP (PX458) from Feng Zhang (Addgene plasmid #48138), containing a GFP expression cassette and the following gRNA-encoding sequence targeting exon 3 of SHOC2: 5'-gRNA-3' GAGCTACATCCAGCGTA ATG, PAM: AGG. GFP-positive cells were sorted by FACS into 96-well plates and single-cell clones were amplified and analysed by western blot to assess SHOC2 protein levels. SHOC2 KO cells were then transduced with lentivirus expressing an empty vector, FLAG-SHOC2 WT or the FLAG-SHOC2D175N, under puromycin selection.

siRNA experiments were performed with a pool of 2 oligos at final concentration of 20 nM. Oligos were mixed with optimum and RNAiMax (ThermoFisher) and added to cells while cells are undergoing attachment. Lysates were harvested 72 h after si transfection.

**Cell proliferation in anchored vs anchorage-independent growth assays.** For growth curves, cells stably expressing shRNAs or CRISPR knockout cells were seeded in 24-well plates and imaged on the IncuCyte (Essen BioScience). Pictures were taken every 2 h, with each data point a composite of four different images. For anchorage independent growth assays, cells were seeded as 8-replicates in low attachment 384-well plates (Greiner). Plates were read on day 7 post seeding by Alamar Blue after 3 h of incubation. Cell seeding was optimised so all lines maintained linear growth over this time.

**Cell viability assays.** Cells were seeded in 384-well plates (Greiner) and left o/n to adhere. Cell seeding was optimised so all lines maintained linear growth over the time frame of the assay. Specifically H358, H520, H727, SK-MEL-2, H1944 were seeded at 2000 cells per well, MEFs and NL20 500 cells per well, whereas all other lines were seeded at a density of 1000 cells per well. The following day drugs were prepared at 10× concentration as serial dilutions for single inhibitor treatments. Cells are incubated in the presence of the drug for 96 h. Cell viability was determined using Cell Titer Glo (Promega) by incubation with the cells for 10 mins. Cell viability was determined by normalizing inhibitor-treated samples to DMSO controls. Alternatively cells were seeded for colony assays in 6-wells at very low confluence, incubated in the presence of drug for 96 h before adding fresh media and staining with crystal violet 7-days after removal of the drug.

**Flow cytometry.** FACS analysis was used as an additional means to quantitate apoptotic, dead and live cell fractions. Cells were seeded, left to adhere o/n and treated with inhibitor the following day. After a 48 h incubation period, cells were harvested and stained with Annexin V-FITC and PI to detect apoptotic cells.

**RBD-RAS pull downs/immunoprecipitation (IP) and western blot.** Levels of active GTP-loaded RAS were determined by GST-RAF-1-RBD pull-down assay. GSTRAF1-RBD fusion proteins 1–149 (Addgene – 13338) were incubated with glutathione beads for 1 h rotating at 4 °C, before extensive washing in PBS-M lysis buffer to remove non-bound protein. Cells for RBD-RAS pull downs were lysed with PBS-M lysis buffer (PBS pH 7.4, 1% w/v Triton X-100, 5 mM MgCl$_2$, 0.1 mM MnCl$_2$), 1 mM DTT, Protease inhibitor cocktail (Roche) and Phosphatase inhibitor

solution (10 mM NaF, 2 mM Na$_3$VO$_4$, 2 mM Na$_4$P$_2$O$_7$, 2 mM β-glycerophosphate). Lysates were incubated for 1 h with the GST-RBD beads rotating at 4 °C before extensive washing in PBS-M lysis buffer to remove non-bound protein.

For endogenous RAF IPs cells were lysed with PBS-E lysis buffer (PBS/1% Triton X-100/1 mM EDTA/ Protease inhibitor cocktail (Roche) and Phosphatase inhibitor solution (10 mM NaF, 2 mM Na$_3$VO$_4$, 2 mM Na$_4$P$_2$O$_7$, 2 mM β-glycerophosphate). Endogenous IPs were performed using combination of the appropriate antibody and Protein A/G beads (Sigma). Lysates were incubated for 6 h rotating at 4 °C before extensive washing in PBS-E lysis buffer to remove non-bound protein.

Immunoprecipitates were drained and resuspended in NuPAGE LDS sample buffer (Life Technologies). Samples were run on western blot for downstream analysis. Uncropped versions of the most important immunoblots are shown in Supplementary Fig. 9–12.

**Xenografts.** A427 KRAS-mutant NSCLC cells ($2.5 \times 10^6$ cells) - & + shSHOC2 were injected subcutaneously into both flanks of 6–8week old female athymic nude mice. For inhibitor experiments once tumours were established (200 mm$^3$), animals (5 per group) were treated with vehicle (4-hydroxypropyl methylcellulose), or Trametinib resuspended in vehicle (0.4 mg/kg daily) for 28-days (No treatment breaks). Tumours were measured twice weekly by digital callipers and mice were weighed weekly for adverse effect to treatments. Tumour volume was calculated using the following formula: tumour volume = (D × d2)/2, in which D and d refer to the long and short tumour diameter, respectively. Animals were culled in accordance with licence restrictions.

**In vivo bioluminescence imaging.** A427 Parental and SHOC2 KO NSCLC luciferase expressing cells ($2.5*10^6$) were injected subcutaneously into the flanks, or, into the lateral tail vein of 6-week old female SCID/ Beige mice (Charles River). 10-days post injections mice were subject to Intraparietal injection with 150 mg/kg of D-Luciferin (GoldBio) dissolved in DPBS. Bioluminescence imaging (BLI) were acquired after 10 min of luciferin injection at a 5-second exposure using the IVIS Lumina. Photons per second were quantified using the IVIS software.

**Generation of SHOC2 KO and SHOC2$^{D175N}$ KI mouse models.** SHOC2 mice were generated by Taconic Artemis. SHOC2 KO model was generated by the insertion of Lox P sites into exon 4 of endogenous SHOC2. For the generation of the SHOC2$^{D175N}$ knockin (KI) mouse model we employed a 'minigene' strategy where the wild-type SHOC2 allele is expressed in a cDNA configuration with a Flag-tag at the N-terminus under the control of the endogenous promoter. The wild-type SHOC2 cDNA sequence is deleted after cre-mediated recombination and replaced by the mutant SHOC2$^{D175N}$ allele containing a Myc-tag.

For SHOC2 KO and KI ERT2 models 6-week old mice are treated with 80 mg/kg tamoxifen treatment in corn oil (Sigma) by oral gavage for 10-days in 2, 5-day treatment windows with a week break in between.

**Lung tumour model.** Mixed-gender, 6- to 12-week old KRAS$^{G12D}$;p53$^{R172H}$; SHOC2$^{wt/wt}$, SHOC2$^{fl/wt}$ or SHOC2$^{fl/fl}$ mice were intranasally infected with a single dose of $2 \times 10^7$ pfu Ad-Cre (University of Iowa Vector Core) to induce tumours. The generation of KRAS$^{G12D}$;p53$^{R172H}$ has been described previously[36,76].

Lungs were isolated at six months post AdenoCre infection. Tumour sections were fixed in 10% formalin (Sigma) o/n before paraffin processing and fractioning. Fractions were stained for H&E. Burden was quantified by determining the total percentage of lung fraction that was tumour at six months. All histopathological analyses were performed blind by an experienced pathologist (M A E-B.).

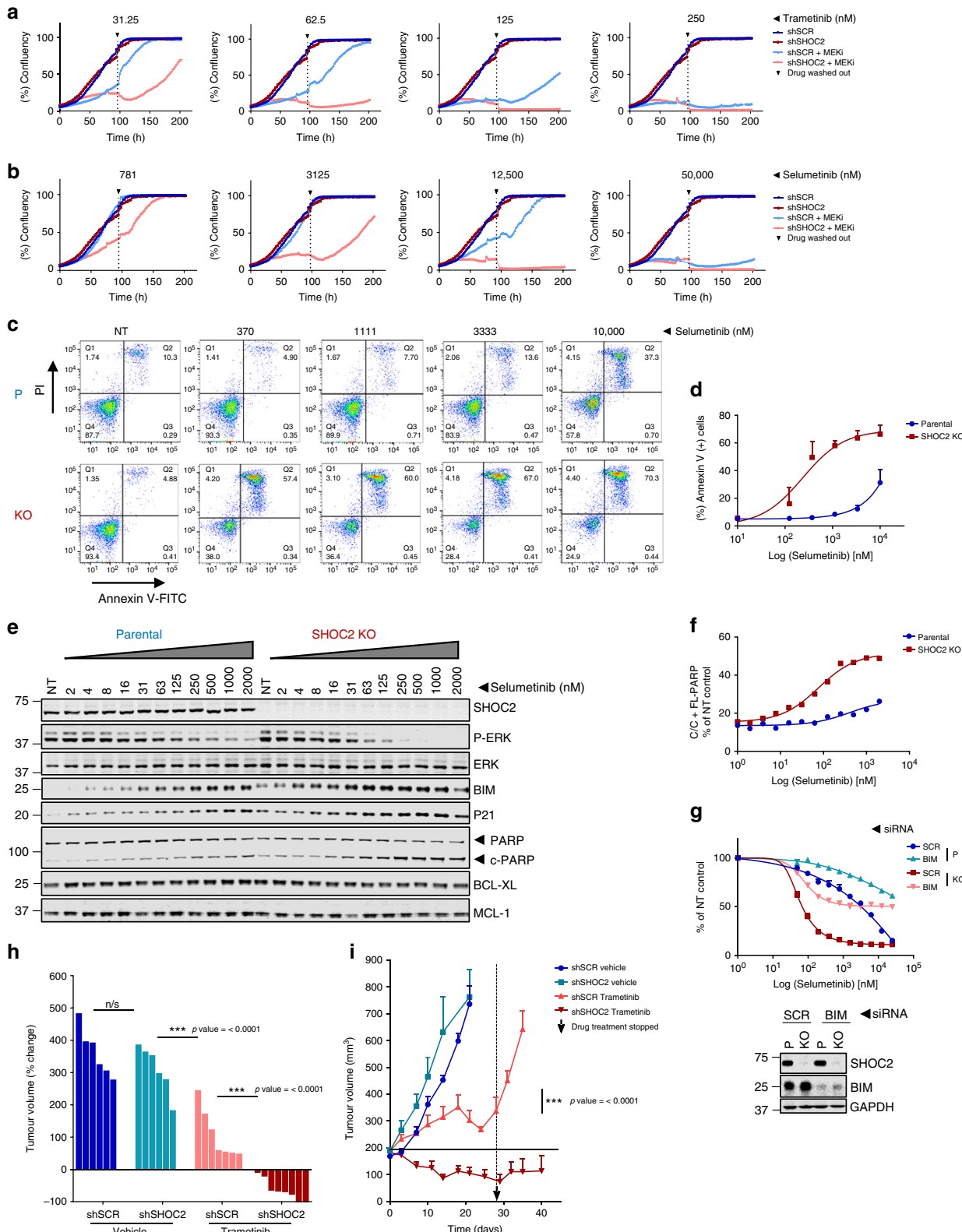

Recombination efficiency of the SHOC2 allele was tested for by PCR in the largest lung tumour nodules from each KRAS[G12D];p53[R172H];SHOC2[fl/fl] mouse to look for 'escapers' (max 2 nodules per animal – where isolation of the nodule from surrounding tumour was easily permissible).

**Mouse embryonic fibroblasts**. Mouse Embryonic Fibroblasts (MEFs) were isolated from R26CreER[T2] SHOC2[fl/wt] mice at p13.5 and plated under standard culture conditions. MEFs were immortalised with FB-E6 and transformed with LXSP3 KRAS[G12V] SHOC2. MEFs were treated with 1 μg/ml 4-OHT (Sigma) for 7-days for to induce Cre-recombination for SHOC2 deletion.

**Animal husbandry**. All mice were maintained in individually ventilated cages (IVCs). Athymic nude mice received autoclaved food, water and bedding according to institutional guidelines. All animal experiments were conducted under an appropriate UK project license in accordance with the regulations of UK home office for animal welfare according to ASPA (animal scientific procedure act 1986).

**Statistical analysis**. Data are presented as mean ± SD unless otherwise stated. Significance was determined with GraphPad Prism 7 software using the Student's $t$-test where $*p < 0.05$, $**p < 0.01$ or $***p < 0.001$.

**Fig. 6** Combined genetic inhibition of SHOC2 and MEKi treatment promote apoptosis in RAS-mutant cells. **a–b** SHOC2 deletion lowers the concentration of MEKi required to prevent re-growth of A427 cells after inhibitor withdrawal. Incucyte growth curves of Parental or SHOC2 KO A427 cells grown in the presence of a single addition of the indicated concentrations of Trametinib (**a**) or Selumetinib (**b**). After 96 h the inhibitor was washed out and cell growth measured for an additional 4 days by incucyte imaging. **c** SHOC2 deletion lowers the concentration of the MEKi required to induce apoptosis in H358 cells. Parental or SHOC2 KO cells were treated with the indicated concentrations of Selumetinib for 48 h and analysed by FACS after PI and Annexin V staining. Representative profiles from $n = 2$. **d** Quantification of (**c**) (mean ± SD)($n = 2$). **e** Immunoblot analysis of lysates of H358 cells treated with Selumetinib for 12 h. **f** Quantification of cleaved PARP in (**e**). **g** BIM knockdown diminishes the sensitisation of H358 SHOC2 KO cells to Selumetinib. H358 P and SHOC2 KO cells transfected with SCR or BIM siRNAs were treated with Selumetinib on Day 2 post transfection and cell viability assays performed 4 days later. **h–i** Combined genetic inhibition of SHOC2 and MEKi treatment promote tumour regressions in A427 xenografts. $2.5*10^6$ shSCR or shSHOC2 A427 cells were injected into the flanks of nude mice and tumours allowed to reach 200 mm³ before treatment with either vehicle or 0.4 mg/kg Trametinib. **h** Change in tumour volume after 28 day treatment is presented as a waterfall plot of individual tumours. Significance is determined using a two tailed $t$-test $*p < 0.05$, $**p < 0.01$ or $***p < 0.001$. **i** Tumour growth over time, Trametinib treatment was stopped after 28 days and tumour growth measured for additional 14 days. $n = 6$ A427 shSCR/ A427 shSHOC2 vehicle, $n = 7$ A427 shSCR Trametinib and $n = 8$ shSHOC2 Trametinib tumours (mean ± SEM)

**Table 1 Antibody list**

| Name | Company | Catalogue number | species | Dilution |
|---|---|---|---|---|
| AKT (pan) | Cell Signaling Technology | 2920 | Mouse | 1:2000 |
| AKT P-S473 | Cell Signaling Technology | 4060 | Rabbit | 1:2000 |
| ARAF | Santa Cruz | sc-166771 | Mouse | 1:1000 |
| ARAF | Santa Cruz | sc-408 | Rabbit | 1:1000 |
| β-Actin | Santa Cruz | sc-47778 | Mouse | 1:5000 |
| BCL-XL | Cell Signaling Technology | 2764 | Rabbit | 1:2000 |
| BIM | Cell Signaling Technology | 2933 | Rabbit | 1:2000 |
| BRAF | Santa Cruz | sc-5284 | Mouse | 1:2000 |
| BRAF | Santa Cruz | sc-9002 | Rabbit | 1:2000 |
| BRAF P-T753 | Abcam | ab138399 | Rabbit | 1:500 |
| CRAF | Santa Cruz | sc-7267 | Mouse | 1:1000 |
| CRAF | BD Biosciences | 610152 | Mouse | 1:1000 |
| CRAF | Santa Cruz | sc-133 | Rabbit | 1:1000 |
| CRAF P-S289/296/301 | Cell Signaling Technology | 9431 | Rabbit | 1:1000 |
| CRAF P-259 | Santa Cruz | Sc-101791 | Rabbit | 1:500 |
| CRAF P-S338 | Cell Signaling Technology | 9427 | Rabbit | 1:1000 |
| EGFR | Santa Cruz | sc-373746 | Mouse | 1:1000 |
| EGFR P-T669 | Cell Signaling Technology | 3056 & 8808 | Rabbit | 1:1000 |
| ERK ½ | Cell Signaling Technology | 9102 | Rabbit | 1:1000 |
| ERK ½ | Cell Signaling Technology | 9107 | Mouse | 1:1000 |
| ERK 1/2 P-T202/Y204 | Cell Signaling Technology | 9101 | Rabbit | 1:1000 |
| FLAG | Sigma | F1365 | Mouse | 1:500 |
| GAPDH | Santa Cruz | sc-47724 | Mouse | 1:5000 |
| KRAS | Santa Cruz | sc-30 | Mouse | 1:500 |
| MCL-1 | Cell Signaling Technology | 5453 | Rabbit | 1:1000 |
| MEK1 | Santa Cruz | sc-6250 | Mouse | 1:1000 |
| MEK2 | Santa Cruz | sc-13159 | Mouse | 1:1000 |
| MEK ½ | Cell Signaling Technology | 4694 | Rabbit | 1:1000 |
| MEK 1/2 P-S217/221 | Cell Signaling Technology | 9121 & 9154 | Rabbit | 1:1000 |
| MYC-TAG | Cell Signaling Technology | 9B11 | Mouse | 1:500 |
| p21 Waf1/Cip1 | Cell Signaling Technology | 2947 | Rabbit | 1:1000 |
| Pan-RAS | Santa Cruz | sc-166691 | Mouse | 1:1000 |
| PARP | BD Biosciences | 556494 | Mouse | 1:1000 |
| PARP (cleaved) | Cell Signaling Technology | 9541 | Rabbit | 1:1000 |
| RPS6 P-S235/236 | Santa Cruz | sc-293144 | Mouse | 1:1000 |
| RSK1 | Santa Cruz | sc-231 | Rabbit | 1:1000 |
| RSK2 | Santa Cruz | sc-9986 | Mouse | 1:000 |
| RSK1 P-S380 | Cell Signaling Technology | 11989 | Rabbit | 1:1000 |
| RSK P-T359/ S363 | Cell Signaling Technology | 9344 | Rabbit | 1:1000 |
| YB1 | Santa Cruz | sc-398340 | Mouse | 1:1000 |
| YB1 P-S102 | Cell Signaling Technology | 2900 | Rabbit | 1:1000 |

**Sequences**. siRNA sequences - Stealth RNAi Negative Control Medium GC Duplex (Life Technologies) was used as control oligo. All other siRNAs were from Qiagen or Life Technologies. SHOC2–1 sense 5′–3′ GCUGCGGAUGCUUGAUUUA antisense 5′–3′ AUUUAGUUCGUAGGCGUCG/ SHOC2–2 sense 5′–3′ GAACUUGGACCAGUAUGGUAGAAUU antisense 5′–3′ CUUGAACCU GGUCAUACCAUCUUAA/ BRAF-1 Sense 5′–3′ AAAGCUGCUUUUCCAGGG UUU antisense 5′3′ AAACCCUGGAAAAGCAGCUUU/ BRAF-2 sense 5′–3′ AA AGAAUUGGAUCUGGAUCAU antisense 5′–3′ AUGAUCCAGAUCCAAUUC UUU CRAF-1 sense 5′–3′ AAGCACGCUUAGAUUGGAAUA antisense 5′–3′ UAUUCCAAUCUAAGCGUGCUU/ CRAF-2 sense 5′–3′ GGAUGUUGAUGGU AGUACA antisense 5′–3′ UGUACUACCAUCAACAUCC/ ARAF-1 5′–3′ sense CCGACUCAUCAAGGGACGAAA antisense 5′–3′ GGCUGAGUAGUUCCCUG CUUU/

shRNA sequences - Clones were obtained from Thermo Scientific Scramble (non-silencing) sense 5′–3′CTCTCGCTTGGGCGAGAGTAAG antisense 5′–3′CTTAC TCTCGCCCAAGCGAGAG/ SHOC2–1 sense 5′–3′ CTGCTGAAATTGGTGAATT antisense 5′–3′ GACGACTTTAACCACTTAA / SHOC2–3 sense 5′–3′ TCTATTC TTTGTAATTACC antisense 5′–3′ AGATAAGAAACATTAATGG.

**Primer sequences**. Kras – WT Forward – 5′–3′ GTCTTTCCCCAGCACAGTGC/ MT Forward 5′–3′ AGCTAGCCACCATGGCTTGAGTAAGTCTGCA/ Common Reverse – CTCTTGCCTACGCCACCAGCTC.

P53 – WT Forward - 5′–3′ TTACACATCCAGCCTCTGTGG/ MT Forward - 5′–3′ AGCTAGCCACCATGGCTTGAGTAAGTCTGCA/ Common Reverse - 5′–3′ CTTGGAGACATAGCCACACTG

SHOC2 conditional KI D175N – 5′–3′ CCATGGACTACAAGGACGACG/ TGATTGTGAGCTACATCCAGGG

SHOC2 conditional KO – 5′–3′ AAACCAGAATGATAGCCAAGCT/ TTGA TAATCCTGCATTAATGGG

SHOC2 WT – 5′–3′ AGTGAAGCTTGAGTCACCATGAG/ GCCGTTTGA TGGTATTGTCG

**Constructs**. pSpCas9(BB)-2A-GFP (PX458) was a gift from Feng Zhang (Addgene plasmid # 48138).

Raf-1 GST RBD 1–149 was a gift from Channing Der (Addgene plasmid # 13338). pBABE-FLAG-LKB1 was a gift from Lewis Cantley (Addgene plasmid # 8592). pLentiX2-PURO-shLKB1-Ms was a gift from Reuben Shaw (Addgene plasmid # 61231).

**Inhibitors**. Pimasertib (AS-703026) (S1475), PD0325901 (S1026), LY3214996 (S8534), LY3009120 (S7842), Ulixertinib (7854), SCH772984 (S7101), OSI-906 (Linsitinib) (S1091), A66 (S2636), GDC-0941 (Pictilisib) (S1065), NVP-AEW541 (S1034), Erlotinib (S7786), Ponatinib (AP24534) (S1490), Afatanib (S1011), MK2206 (S1078), GDC-0068 (Ipatasertib) (S2808), BGJ398 (S2183), Crizotinib (S1068), RAF265 (S2161), AZD628 (S2746), LY3009120 (S7842) were purchased from SelleckChem. Trametinib (GSK1120212) (871700–17–3), AZD6244 (Selumetinib) (606143–52–6) were purchased from Generon. Drugs for in vitro studies were dissolved in DMSO and stored at −20 °C. Drugs for in vivo studies were made fresh by resuspending compound in 0.5% 4-hydroxypropyl methylcellulose, 0.2% Tween-80 (Sigma).

**Antibody list**. BRAF P-S365 was generated by immunisation of rabbits with a phoshpo-peptide corresponding to the appropriate region of BRAF (Epitomics/ Abcam). SHOC2 antibody was generated as described[26]. All other antibodies were sourced and used as indicated in Table 1.

**Reporting summary**. Further information on research design is available in the Nature Research Reporting Summary linked to this article.

## Data availability

All data supporting the findings of the current study are available within the article and its Supplementary Information files or from the corresponding author upon request.

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

## Acknowledgements

We wish to thank Henning Walczak and Silvia von Karstedt (University College London) for supplying us with the KRAS^G12D;Trp53^R172H mouse models. We also would like to thank Mani Venkatesan, Lucia Cottone (University College London) and Romain Baer (Francis Crick Institute) for materials and reagents. In addition we would like to acknowledge Lorraine Lawrence (NHLI, Imperial College, London, UK) and Antonella Montinaro (University College London) for assistance with processing and scoring of histology samples. We would also like to thank Frank McCormick (University of California San Francisco), Asim Khwaja and Benoit Bilanges (University College London), for manuscript review and feedback. The contribution of G.J., I.B.R., I.A.Z. was supported by Cancer Research UK grants. S.S. is sponsored by the 'The Republic of Turkey Ministry of National Education'. We are grateful to Rosetrees Trust for supporting this project (M190-F1).

## Author contributions

G.J. and P.R.V. conceived the project, designed the experiments and co-wrote the manuscript. G.J., I.B.R., S.S., A.S., I.A.Z., W.L., L,C,Y., N.H., R.E.H. supported laboratory experiments, and analysis of results. M.A.E-B. analysed histology samples derived from in vivo work. J.D. and M.M.-A provided critical review and feedback on the project, as well as supporting the development of in vitro systems. Project administration and supervision was carried out by P.R.V.

## Additional information

**Competing interests:** The authors declare no competing interests.

