## [Peer Review File · Nature Communications]

Reviewers' Comments:

Reviewer #1:

Remarks to the Author:

Summary: Despite the ubiquity of mutations in KRAS in non-small cell lung cancer (NSCLC), no effective anti-RAS therapies have transitioned into the clinic for NSCLC. Therefore, many groups have focused on the identification of pharmacologically accessible signaling pathways downstream of oncogenic RAS. In agreement with the requirement for MAPK pathway activation downstream of oncogenic RAS, small molecule inhibitors of each node have been approved or are under evaluation for therapeutic targeting in multiple cancers. Despite overwhelming evidence that targeting the RAF-MEK-ERK axis is efficacious in targeting hyperactivated MAPK pathway tumors, the responses are not durable due in large part to resistance. As described by Jones et al. the protein phosphatase SHOC2 is an instrumental regulator of RAS-mediated RAF kinase activation via dephosphorylation of a negative regulatory site within the kinases that prevents their dimerization. In the paper under review, Jones et al. seek to systematically interrogate the contribution of the SHOC2 phosphatase complex in KRAS-driven lung adenocarcinoma through the utilization of both a genetically engineered mouse model and human cancer cell lines to genetically ablate *Shoc2* or express an inactive mutant form of SHOC2, pharmacologically inhibit SHOC2 with a small molecule inhibitor, and test synergistic combinations for increasing the efficacy of targeting SHOC2. In simplistic but elegant experiments, the authors generate two novel strategies to conditionally inactivate SHOC2 function either through complete genetic deletion or expression of a minigene that encodes the SHOC2 D175N point mutant that were utilized to test the impact of SHOC2 dysfunction specifically in tumor tissue or in the entire animal. In addition, the authors corroborated their findings in human lung adenocarcinoma cell lines and established a potential model for how RAF regulation and subsequent MAPK pathway activation are altered in the context of SHOC2 inhibition. Together, these data support the notion that SHOC2 could be a therapeutic target in NSCLC. While the paper is experimentally solid and the tools are new to the field, the concepts lack some novelty without an actual SHOC2 inhibitor that could be mitigated by weakening the language that indicates that knockdown or knockout experiments or inhibition. In addition, there are additional experiments that would improve the manuscript as submitted that warrant revision.

Questions:

1. The authors provide convincing data that genetic loss of SHOC2 or expression of a SHOC2 mutant decrease KRAS-driven lung adenocarcinoma formation. However, the authors do not include the Kaplan Meier survival data for the SHOC2 knock in mice in the KRAS, p53 model. The authors should include these data to determine whether the knockin has a different phenotype in this model.
2. While the authors clearly show that knockdown of SHOC2 in a panel of human lung adenocarcinoma cell lines increases phosphorylation of BRAF P-S365, it is well appreciated that oncogenic KRAS utilizes the CRAF-BRAF heterodimer for MAPK signaling. The authors should repeat the experiments in Figure 2A and 2I to interrogate phosphorylation of CRAF P-S259 in order to determine whether SHOC2 loss also increases CRAF P-259 and contributes to the anti-tumorigenic properties (xenograft growth and 3D growth).
3. The authors suggest that the A427 cell is 'resistant' to SHOC2 genetic knockdown, while the cells when injected into tail vein form less tumors in the lung. The authors see differential phenotypes in the other cell lines that have phenotypes in 3D growth and xenograft experiments. While the authors attempt to address this with signaling experiments, each of the cell lines responds similarly in 2D and 3D in response to SHOC2 knockdown. The authors need to mechanistically address why these differential phenotypes occur. Is it based on a cell line mutational status or other signaling pathways? One of the striking differences in the 2D A427 line studies in the absence of SHOC2 is increased phosphorylation of AKT. Does this explain the differences?

4. The authors indicated that "However, SHOC2 inhibition lowered IC50 values to MEKis to a similar extent in both adhered and suspension culture conditions (data not shown), and therefore viability assays were performed in subsequent studies under adhered conditions." These data are critical to the interpretation of the results in subsequent figures on the combination of SHOC2 knockdown and MEK1/2 inhibitor efficacy in 2D. These data not shown should be included in the manuscript.

5. The authors do a well-designed genetic experiment in Figure 3I-J and Supp Figure 2B in which they knockout endogenous SHOC2 and overexpress phosphatase insensitive mutants of either ARAF, BRAF, or CRAF and test both MAPK signaling or sensitivity to MEK1/2 inhibitors. However, these experiments are done in the SHOC2 resistant cell line A427 in Figure 3I-J. What explains the enhanced sensitivity now when the signaling changes were the same in both of these lines?

Reviewer #2:

Remarks to the Author:

The manuscript by Jones et al, reports on a potential role of SHOC2 in KRAS mutant NSCLC progression as well as in preventing durable ERK pathway suppression by MEK inhibitors. The data provide a rationale for combinatorial targeting of SHOC2 and MEK in KRAS and EGFR mutant NSCLC tumors. The authors identify a subset of NSCLC cells in which SHOC2 downregulation synergizes with MEK inhibition *ex vivo* and *in vivo* and observe consistently higher sensitivity to SHOC2 downregulation of tumor cells when grown in 3D compared to 2D conditions.

The manuscript includes a considerable amount of experimental work many of the observations reported here are interesting and potentially therapeutically important, however certain issues relating to underlying mechanisms need to be addressed:

1. A major concern on the proposed mechanism(s), is that the effect of SHOC2 KO on ERK pathway activity may be not due to a direct effect on RAF, but rather a consequence of inhibition of upstream RAS activity, which in turn results in suppression of RAF dimerization and activity. Have the authors monitored RAS activity (as determined by RAS-pull down assays) and found it to be similar between P and SHOC2 KO cells in 2A, 2I? Also, CRAF is important in driving ERK pathway activation in NSCLC - what is the effect of SHOC2 downregulation on S259CRAF and S338CRAF? Further, in Figure 4A, is feedback-induction of RAS activity and S338CRAF similar between P and SHOC2 KO cells upon treatment with selumetinib?

2. The effect of SHOC2 KO may vary, depending on the RAS isoforms and/or mutation. The authors previously reported that SHOC2 functions selectively downstream of MRAS (Mol. Cell 2006), in this manuscript however they propose that it affects ERK pathway activation downstream of KRAS. Have the authors looked at potential synergy between SHOC2 KO and MEK inhibitor treatment in cells with a KRAS mutation other than G12, such as Calu6 (Q61K), in other KRAS-mutant tumors such as colorectal and pancreatic or in NRAS-mutant tumors, as in melanoma?

3. While suppression of ERK pathway upon SHOC2 downregulation is more or less consistent in NSCLC (especially under 3D culture conditions), the effect of SHOC2 downregulation on cell growth is quite variable (only 3 out of 6 cell lines were relatively sensitive to SHOC2 KO) and no rationale or mechanism is provided to explain the observed differences. This raises questions on both the generality of the findings, as well as on the proposed clinical implications. Is there another function of SHOC2 outside ERK pathway activation that may account for the differences? Is it possible that certain RAS mutations are more sensitive to SHOC2 downregulation than others?

4. The experiments shown with the RAF inhibitor LY3009120 are confusing. RAF inhibitor-induced RAF dimerization has been shown previously by a number of groups to require RAS activity. The

authors reached a different conclusion presumably because they used a relatively low concentration of LY3009120 (2.5uM), in which ERK pathway is not inhibited deeply enough to show feedback-activation of RAS, as in the selumetinib or trametinib-treated cells. Had the authors used a higher concentration (7-10uM) of LY3009120, they may have observed differences in the ERK pathway rebound and in CRAF/BRAF dimerization between P and SHOC2 KO cells with the pan-RAF inhibitor as well.

5. It is surprising that an ERK inhibitor (Ulixetinib) does not synergize with SHOC2 KO, as is the case with MEK inhibitors. How is this consistent with the authors proposed model of the role of SHOC2 in regulating ERK pathway activity?

6. What is the basis for the consistent increase in pAKT upon SHOC2 KO? Could that represent a potential liability of the proposed therapeutic strategy?

Minor:

I have not been able to locate where Figure S2C is cited in the manuscript.

Reviewer #3:

Remarks to the Author:

In this manuscript, Jones et al. discover a novel role for SHOC2 in promoting KRAS-driven anchorage independent growth, and synergy between SHOC2 depletion and MEK inhibition in KRAS and EGFR mutant lung cancer cell lines. First, they demonstrate impairment of lung tumor formation in *Kras*;p53 and *Kras* genetically engineered mouse models following SHOC2 deletion, and demonstrate that systemic SHOC2 ablation in adult mice is well tolerated over an 8 week period. Next, they uncover a specific impact of SHOC2 knockout on 3D cell growth and tumor xenograft formation or metastatic lung colonization in several KRAS mutant cell lines. More impressively they uncover potent synergy of SHOC2 deletion with MEK inhibition in multiple KRAS and EGFR mutant cell lines. Mechanistically they observe that SHOC2 loss prevents rebound pBRAf, pMEK, and pERK reactivation, and that this relates to disruption of MEKi induced RAF dimer formation. Finally, they demonstrate that combined suppression of SHOC2 and MEK inhibition induces apoptosis involving BIM, with impressive synergy in vivo in an A427 xenograft model.

Overall this is a very nice study that uncovers a novel target that synergizes with MEKi. I only have a few concerns that should be addressed prior to publication.

1. My major concern is the claim that SHOC2 deletion is well tolerated systemically, and that this therapeutic window is highlighted throughout the discussion. The authors are proposing MEK inhibitor combination studies, and no evidence is provided that SHOC2 deletion + MEK inhibition is tolerated systemically. Have the authors tried treating their SHOC2 KO mice with MEK inhibitors to see whether or not there is significantly increased toxicity?

2. Along these same lines the MEF experiment in Figure 4F does show that SHOC2 deletion impairs P-ERK reactivation in the empty vector cells relative to KRAS G12V, except for the 30 min timepoint. Quantitation of P-ERK to total ERK signal would help to highlight the difference they are claiming in the absence or presence of KRAS G12V (though they do show that MEFs tolerate SHOC2 KO + MEKi relative to KRAS mutant cells in 3E). As an alternative to treating the SHOC2 KO mice with MEK inhibitors, at a minimum it would be important to include additional data that human fibroblasts and/or AALE cells are less sensitive to the combination compared to lung cancer cells. And regardless the discussion should have appropriate caveats that the therapeutic window of combined systemic SHOC2 and MEK inhibition is unclear.

3. The authors compare 2D and 3D effects of SHOC2 KO but only 2D effects of the MEKi combination. Does MEKi also impair 3D spheroid growth of KRAS mutant cell lines such as A549

and A427, which were resistant to SHOC2 KO?

Minor Concerns

1. Figure 4G is referred to before Figure 4F in the text, figures should be called out in order.
2. Unclear why H522 data not shown? Having another KRAS WT example where pERK feedback is minimally suppressed by SHOC2 KO would strengthen the story, especially since the MEF data in Fig 4F is not entirely convincing.

Reviewers' comments:

We would like to thank the reviewers for their constructive criticism and thoughtful suggestions. As a result of their positive feedback we have now considerably expanded the observations and breadth of this study.

Reviewer #1 (Remarks to the Author):

Summary: Despite the ubiquity of mutations in KRAS in non-small cell lung cancer (NSCLC), no effective anti-RAS therapies have transitioned into the clinic for NSCLC. Therefore, many groups have focused on the identification of pharmacologically accessible signaling pathways downstream of oncogenic RAS. In agreement with the requirement for MAPK pathway activation downstream of oncogenic RAS, small molecule inhibitors of each node have been approved or are under evaluation for therapeutic targeting in multiple cancers. Despite overwhelming evidence that targeting the RAF-MEK-ERK axis is efficacious in targeting hyperactivated MAPK pathway tumors, the responses are not durable due in large part to resistance. As described by Jones et al. the protein phosphatase SHOC2 is an instrumental regulator of RAS-mediated RAF kinase activation via dephosphorylation of a negative regulatory site within the kinases that prevents their dimerization. In the paper under review, Jones et al. seek to systematically interrogate the contribution of the SHOC2 phosphatase complex in KRAS-driven lung adenocarcinoma through the utilization of both a genetically engineered mouse model and human cancer cell lines to genetically ablate Shoc2 or express an inactive mutant form of SHOC2, pharmacologically inhibit SHOC2 with a small molecule inhibitor, and test synergistic combinations for increasing the efficacy of targeting SHOC2. In simplistic but elegant experiments, the authors generate two novel strategies to conditionally inactive SHOC2 function either through complete genetic deletion or expression of a minigene that encodes the SHOC2 D175N point mutant that were utilized to test the impact of SHOC2 dysfunction specifically in tumor tissue or in the entire animal. In addition, the authors corroborated their findings in human lung adenocarcinoma cell lines and established a potential model for how RAF regulation and subsequent MAPK pathway activation are altered in the context of SHOC2 inhibition. Together, these data support the notion that SHOC2 could be a therapeutic target in NSCLC. While the paper is experimentally solid and the tools are new to the field, the concepts lack some novelty without an actual SHOC2 inhibitor that **could be mitigated by weakening the language that indicates that knockdown or knockout experiments or inhibition**. In addition, there are additional experiments that would improve the manuscript as submitted that warrant revision.

We have changed the language to reflect this concern and we have now inserted throughout the manuscript the term '*genetic*' when referring to 'SHOC2 inhibition' (i.e. '*genetic SHOC inhibition*'). Furthermore we now use routinely use '*deletion/depletion*' (instead of inhibition) to indicate whether the genetic inhibition is shRNA (knockdown/depletion) or CRISPR/ R26CreER^{T2} (knockout/deletion).

Questions:

1. The authors provide convincing data that genetic loss of SHOC2 or expression of a SHOC2 mutant decrease KRAS-driven lung adenocarcinoma formation. However, the authors do not include the Kaplan Meier survival data for the SHOC2 knock in mice in the KRAS, p53 model. The authors should include these data to determine whether the knockin has a different phenotype in this model.

For technical reasons related to its generation and subsequent crosses, experimental mice with KI Shoc2 have always lagged behind relative to the KO mice, with the KRAs/p53/Shoc2KI colony

in particular being the slowest to get going. Because of the high cost associated with this type of animal work and funding-related issues we were forced to make cuts in the size of our mouse colony.

Considering we have not seen any differences between KO and KI in other systems and the severe lag in numbers and timeline of the KRas/p53/Shoc2KI colony, we stopped further breeding of this colony to focus instead on the more advanced KRas/Shoc2KI colony for the initiation model and use our remaining available resources to develop with the Shoc2KI a tumour maintenance model that will be the focus of future studies. At the time of this decision, there were only 3 experimental KRas/p53/Shoc2KI mice generated that were used for survival experiments.

We include below a version of the Kaplan Meir plot that includes these 3 KRas/p53/Shoc2KI mice that clearly strongly suggests the Shoc2KI mice do behave similarly to the KO in the KRas/p53 model (as well as in the Kras-only model). However we are conscious of the fact that n=3 is generally not acceptable for publication, which is why we did not include the data previously.

We are afraid we would be unable to provide more numbers to the curve in a realistic time frame but hope this is enough to satisfy the reviewer on this point. We leave it up to the reviewer and/or editors to decide whether the below plot with n=3 for the KI is acceptable for publication (and we should include this version of the Figure 1F) or whether we leave the current version that does not include KI mice on KRas/p53 model.

2. While the authors clearly show that knockdown of SHOC2 in a panel of human lung adenocarcinoma cell lines increases phosphorylation of BRAF P-S365, it is well appreciated that oncogenic KRAS utilizes the CRAF-BRAF heterodimer for MAPK signaling. The authors should repeat the experiments in Figure 2A and 2I to interrogate phosphorylation of CRAF P-S259 in order to determine whether SHOC2 loss also increases CRAF P-259 and contributes to the anti-tumorigenic properties (xenograft growth and 3D growth).

Whereas we have raised our own P-S365 BRAF antibody that works very well and we have comprehensively validated, we use a commercially available antibody against P-S259 CRAF that is less sensitive and for which we have found batch to batch variations with sensitivity and cross reactivity.

Regardless, we now include a new Figures S1A where we show that although P-S259 CRAF basal levels are less sensitive to SHOC2 KO compared to P-S365 BRAF, EGF-induced S259 CRAF dephosphorylation (but not P-S338 CRAF) is clearly inhibited in the absence of SHOC2, both in RAS-mutant (A549) and RAS-wild type cells (HEK293T).

We also now include panels with P-S259 CRAF levels in Figure 4A and 4D (A549, H358, A427, and HCC4006 cells, with or without MEKi treatment) to show that SHOC2 loss also leads to higher P-259 CRAF levels (as well as P-S365 BRAF levels) which is consistent with our previously published observations that the SHOC2 complex works as a 'S259' phosphatase on all RAF paralogues (Rodriguez-Viciano et al 2006).

3. The authors suggest that the A427 cell is 'resistant' to SHOC2 genetic knockdown, while the cells when injected into tail vein form less tumors in the lung. The authors see differential phenotypes in the other cell lines that have phenotypes in 3D growth and xenograft experiments. While the authors attempt to address this with signaling experiments, each of the cell lines responds similarly in 2D and 3D in response to SHOC2 knockdown. The authors need to mechanistically address why these differential phenotypes occur. Is it based on a cell line mutational status or other signalling pathways? One of the striking differences in the 2D A427 line studies in the absence of SHOC2 is increased phosphorylation of AKT. Does this explain the differences?

The reviewer is right to point out that SHOC2's contribution to ERK signalling in 3D (seen across all RAS mutant cell lines tested) is not sufficient to account for differential phenotypes observed in different experimental contexts in different cell lines. While in the previous manuscript we had indeed speculated in the discussion that these differences could be due to either differential mutational status (e.g. LKB1) and/or increased AKT signalling, we have now devoted considerable effort to directly addressing this important point.

We now include new data in new Figure 2 J-L and new Figure S2 where we have performed knock-down and overexpression studies of LKB1 and myr-AKT in multiple cell lines. Our results indeed suggest that increased AKT activation may be at least partly responsible for 3D growth in the absence of SHOC2 in some of the '3D resistant' cell lines.

In the results section to accompany Figure 2, we now include the following paragraph:

We noted that cell lines resistant to SHOC2 depletion for spheroid growth (A421, H460, A549) have inactivating mutations in the LKB1/STK11 tumour suppressor gene (Table S1) as well as retaining higher AKT phosphorylation levels in suspension (Figure 2I), suggesting a possible role for LKB1 and/or PI3K/AKT signalling in overcoming a requirement for SHOC2 for anchorage-independent growth. LKB1 re-expression in A427 LKB1 null cells failed to render them sensitive to SHOC2 depletion (Figure S2D), whereas LKB1 knockdown failed to overcome SHOC2 requirement for spheroid growth in H358 '3D sensitive' cells (Figure S2E). Thus, LKB1 status alone does not appear to determine sensitivity to SHOC2 for 3D growth. On the other hand, ectopic expression of membrane associated, constitutively activate AKT (Myr-AKT1), had no effect on anchorage dependent or independent growth in 'SHOC2 3D resistant' A549 cell lines but fully rescued spheroid growth in SHOC2 3D sensitive' H358 and H1792 SHOC2 KO cell lines (Figure 1J-L and S2F-I). Thus, at least in some contexts, increased AKT signalling may help to overcome SHOC2 requirement for 3D growth in RAS-mutant cells.

4. The authors indicated that "However, SHOC2 inhibition lowered IC50 values to MEKis to a similar extent in both adhered and suspension culture conditions (data not shown), and therefore viability assays were performed in subsequent studies under adhered conditions." These data are critical to the interpretation of the results in subsequent figures on the combination of SHOC2 knockdown and MEK1/2 inhibitor efficacy in 2D. These data not shown should be included in the manuscript.

A new Figure S3B has now been included to illustrate this point that SHOC2 deletion sensitizes cells to MEKi under both adhered and suspension culture conditions.

5. The authors do a well-designed genetic experiment in Figure 3I-J and Supp Figure 2B in which they knockout endogenous SHOC2 and overexpress phosphatase insensitive mutants of either ARAF, BRAF, or CRAF and test both MAPK signaling or sensitivity to MEK1/2 inhibitors. However, these experiments are done in the SHOC2 resistant cell line A427 in Figure 3I-J. What explains the enhanced sensitivity now when the signaling changes were the same in both of these lines?

We believe this point arises from a confusion fostered by our previous use of the term 'SHOC2 resistant/sensitive' in the results accompanying Figure 2 where 'sensitivity/resistance' refers exclusively to 3D growth. Regardless of their behaviour in 3D/spheroid assays (see also point 3 above), all RAS-mutant cell lines tested (except H727 that is also ARAF mutant) are sensitized to MEKi upon SHOC2 deletion/depletion. To avoid this misunderstanding, we now insert the term '3D' when referring to 'SHOC2 3D sensitive' or 'SHOC2 3D resistant' cell lines to specify in the text accompanying Figure 2 that sensitivity/resistance refers to 3D growth only.

Reviewer #2 (Remarks to the Author):

The manuscript by Jones et al, reports on a potential role of SHOC2 in KRAS mutant NSCLC progression as well as in preventing durable ERK pathway suppression by MEK inhibitors. The data provide a rationale for combinatorial targeting of SHOC2 and MEK in KRAS and EGFR mutant NSCLC tumors. The authors identify a subset of NSCLC cells in which SHOC2 **downregulation** synergizes with MEK inhibition ex vivo and in vivo and observe consistently higher sensitivity to SHOC2 downregulation of tumor cells when grown in 3D compared to 2D conditions.

The manuscript includes a considerable amount of experimental work many of the observations reported here are interesting and potentially therapeutically important, however certain issues relating to underlying mechanisms need to be addressed:

1. A major concern on the proposed mechanism(s), is that the effect of SHOC2 KO on ERK pathway activity may be not due to a direct effect on RAF, but rather a consequence of inhibition of upstream RAS activity, which in turn results in suppression of RAF dimerization and activity. Have the authors monitored RAS activity (as determined by RAS-pull down assays) and found it to be similar between P and SHOC2 KO cells in 2A, 2I? Also, CRAF is important in driving ERK pathway activation in NSCLC - what is the effect of SHOC2 downregulation on S259CRAF and S338CRAF?

We now include a new Figure S1A, where we have performed RAF-RBD pull-down assays to monitor RAS-GTP loading. We show that RAS activation is clearly not impaired in the absence of SHOC2. In fact, a modest increase in RAS-GTP loading is suggested by some experiments, consistent with relief of inhibitory feedbacks (this is more comprehensively characterized in an expanded panel of cell lines in another manuscript in preparation).

Because in the RAS-mutant cell lines we have tested we observe high basal levels of RAS-GTP (and S338 CRAF phosphorylation) that are not significantly stimulated by EGF, we also include in Figure S1A, data for HEK293T RAS-wild type cells from an experiment performed in parallel with A549 cells, as a 'positive control' for RAS activation/S338 CRAF phosphorylation by EGF to rule out that lack of regulation by EGF in RAS-mutant cells is due to experimental considerations.

We also now include in Figure S1A and Figures 4A and 4D panels for phospho-S259 CRAF and phospho-S338 CRAF levels that show that SHOC2 KO/KD impairs S259 CRAF (as well as S365 BRAF) dephosphorylation but not S338 CRAF phosphorylation.

Further, in Figure 4A, is feedback-induction of RAS activity and S338CRAF similar between P and SHOC2 KO cells upon treatment with selumetinib?

In Figures 4A and 4D we now include panels for phospho-S338 CRAF levels and in Figure 4D RAF-RBD pull-downs to monitor RAS activity. As seen in the case of EGF treatment, we do not detect any significant increase in RAS-GTP or P-S338 CRAF levels upon Selumetinib treatment, likely because of the high basal levels of KRAS-GTP in these KRAS mutant cell lines.

2. The effect of SHOC2 KO may vary, depending on the RAS isoforms and/or mutation. The authors previously reported that SHOC2 functions selectively downstream of MRAS (Mol. Cell 2006), in this manuscript however they propose that it affects ERK pathway activation downstream of KRAS. Have the authors looked at potential synergy between SHOC2 KO and MEK inhibitor treatment in cells with a KRAS mutation other than G12, such as Calu6 (Q61K), in other KRAS-mutant tumors such as colorectal and pancreatic or in NRAS-mutant tumors, as in melanoma?

We have now also expended considerable effort to address this important point and in addition to H460 cells (KRAS Q61H, NSCLC,) we have now generated SHOC2 KO/KD cells for H1944 (KRAS G13D, NSCLC), HCT-116 (KRAS G13D, Colorectal), MDA-MB-231 (KRAS G13D, TNBC), PATU-8092 (KRAS G12V Pancreatic) and SK-MEL-2 (NRAS Q61R, Melanoma) cells. We show in new Figure 3B and S4C that SHOC2 KO/KD sensitizes in all cases to MEK inhibitors, i.e. sensitization to MEK inhibitors upon SHOC2 ablation is seen across all RAS-mutant cell lines tested, regardless of tumour type of origin and RAS mutation type.

Of note, we had indeed shown that SHOC2 functions downstream to MRAS, but critically, to regulate RAF recruited by H/N/K-RAS and furthermore we have also shown that SHOC2 downregulation inhibited ERK phosphorylation in a panel of KRAS- and NRAS-mutant cells (Rodriguez-Viciana et al, Mol Cell, 2006 Figure 7 and S1).

3. While suppression of ERK pathway upon SHOC2 downregulation is more or less consistent in NSCLC (especially under 3D culture conditions), the effect of SHOC2 downregulation on cell growth is quite variable (only 3 out of 6 cell lines were relatively sensitive to SHOC2 KO) and no rationale or mechanism is provided to explain the observed differences. This raises questions on both the generality of the findings, as well as on the proposed clinical implications. Is there another function of SHOC2 outside ERK pathway activation that may account for the differences?

This is a similar point as raised by Reviewer 1 (point 3) addressed before, and we will repeat much of that answer.

The reviewer is right to point out that SHOC2's contribution to ERK signalling in 3D (seen across all RAS-mutant cell lines tested) is not sufficient to account for differential phenotypes observed in different experimental contexts in some of these cell lines. While in the previous version of the manuscript we had speculated in the discussion that these differences could be due to either differential mutational status (e.g. LKB1) and/or increased AKT signalling, we have now devoted considerable effort to directly address this important point.

We now include new data in new Figure 1 J-L and new Figure S2 where we have performed knock-down and overexpression studies of LKB1 and myr-AKT in multiple cell lines and indeed show that higher basal AKT activity in 3D, may provide a molecular mechanism for anchorage-independent growth in the absence of SHOC2.

In the results section to accompany Figure 2, we now include the following paragraph:

We noted that cell lines resistant to SHOC2 depletion for spheroid growth (A421, H460, A549) have inactivating mutations in the LKB1/STK11 tumour suppressor gene (Table S1) as well as retaining higher AKT phosphorylation levels in suspension (Figure 2I), suggesting a possible role for LKB1 and/or PI3K/AKT signalling in overcoming a requirement for SHOC2 for anchorage-independent growth. LKB1 re-expression in A427 LKB1 null cells failed to render them sensitive to SHOC2 depletion (Figure S2D), whereas LKB1 knockdown failed to overcome SHOC2 requirement for spheroid growth in H358 '3D sensitive' cells (Figure S2E). Thus, LKB1 status alone does not appear to determine sensitivity to SHOC2 for 3D growth. On the other hand, ectopic expression of membrane associated, constitutively

activate AKT (Myr-AKT1), had no effect on anchorage dependent or independent growth in A549 and H1792 'SHOC2 3D resistant' cell lines but fully rescued spheroid growth in H358 and H1792 'SHOC2 3D sensitive' cell lines (Figure 1J-L and S2F-I). Thus, at least in some contexts, increased AKT signalling may help to overcome SHOC2 requirement for 3D growth in RAS-mutant cells.

In the discussion we say:

... Conversely, our data suggests that aberrant signalling by the PI3K/AKT (and/or other pathway(s)) can compensate for loss of SHOC2-dependent ERK-signalling to promote anchorage-independent growth in some contexts. Regardless, SHOC2's contribution to tumorigenic properties in some RAS-mutant human cells lines, as well as to tumour development in a KRAS driven mouse LUAD model suggests targeting SHOC2 in the clinic may have activity as monotherapy against a subset of RAS-mutant cancers.

Is it possible that certain RAS mutations are more sensitive to SHOC2 downregulation than others?

As addressed in point 2 above, we have now generated SHOC2 KO/KD variants for additional RAS-mutant cell lines with KRAS substitutions at positions G12, G13 and Q61, as well as one melanoma NRAS Q61 cell line (Figure 3, S4C) and show that in all cases they are sensitized to MEKi treatment independently of the type of activating RAS mutation or tumour type of origin. We have as yet been unable to find a RAS-mutant cell line that is not sensitized to MEKi by efficient SHOC2 loss (other than H727) but these are ongoing studies.

4. The experiments shown with the RAF inhibitor LY3009120 are confusing. RAF inhibitor-induced RAF dimerization has been shown previously by a number of groups to require RAS activity. The authors reached a different conclusion presumably because they used a relatively low concentration of LY3009120 (2.5uM), in which ERK pathway is not inhibited deeply enough to show feedback-activation of RAS, as in the selumetinib or trametinib-treated cells. Had the authors used a higher concentration (7-10uM) of LY3009120, they may have observed differences in the ERK pathway rebound and in CRAF/BRAF dimerization between P and SHOC2 KO cells with the pan-RAF inhibitor as well.

We believe this point arises from our previous incorrect use of '*RAS-independent RAF dimerization*' within the results section in the previous manuscript: (*...In contrast however, the potent RAS-independent RAF dimerization induced by the RAFi LY3009120^{50, 51} is not affected by loss of SHOC2 (Figure 5B)...*)

Our previous statement stemmed from our referencing of the study by Jin et al (NatCom 2017) that states: '*...RAF inhibitors induce the disruption of intramolecular interactions between the kinase domain and its N-terminal regulatory region independently of RAS activity.*'

We appreciate that although RAF inhibitor-induced RAF dimerization may have a RAS-independent component, RAS activity may still be required for RAF dimerization. We therefore have now removed '*RAS-independent*' when referring to RAF inhibitor-induced dimerization in the current manuscript.

With regards to whether the concentration of LY3009120 used in Figure 5C (2.5µM, which is enough to promote potent SHOC2-independent RAF dimerization) is high enough to efficiently inhibit activity and therefore cause feedback-relief, we note that in Figure 4F, where cells were similarly pre-treated with LY3009120 for **12 hours** before washing out the inhibitor, there is clear signalling rebound at the level of P-MEK and P-ERK upon inhibitor removal, consistent with effects on feedback-relief. Furthermore, we now show in new Figures S5G-H, a dose response of LY3009120 in RAS-mutant H358 cells, to show 2.5µM LY3009120 does inhibit ERK pathway activity (as measured by P-MEK, P-ERK and P-RSK) when cells are **acutely treated for 30 min**.

5. It is surprising that an ERK inhibitor (Ulixetinib) does not synergize with SHOC2 KO, as is the case with MEK inhibitors. How is this consistent with the authors proposed model of the role of SHOC2 in regulating ERK pathway activity?

This is indeed a very intriguing observation. We now include a new Figure 5D, where we measure the response of the ERK inhibitor LY3214996 on a wash-out experiment. We show that, as seen with MEKi, RAF dimerization and MEK rebound phosphorylation by ERKi is SHOC2-dependent. However, in clear contrast to MEKi (but like RAFi), ERK phosphorylation is unaffected by SHOC2 loss. Although the molecular mechanism remains unclear, this observation is likely to be at the heart (at least partly) of the selective sensitization observed to MEKi, but not RAFi or ERKi. Future studies beyond the scope of this manuscript should shed light on these surprising differential responses depending on the node of the RAF-MEK-ERK cascade that is targeted.

6. What is the basis for the consistent increase in pAKT upon SHOC2 KO? Could that represent a potential liability of the proposed therapeutic strategy?

We observe that phosphorylation of ERK feedback sites on T669 EGFR is strongly impaired in the absence of SHOC2 (e.g, Figure 6A and unpublished results) providing a possible mechanism to increased P-AKT upon SHOC2 KO. We note however, that inhibitory crosstalk between ERK and PI3K pathways and the resulting increased AKT phosphorylation upon MEKi treatment has been previously reported (e.g. Gan Y, et al. 2010, Turke AB, et al. 2012). As such, increased P-AKT as a potential liability of anti-SHOC2 therapies is only a reflection of its role within the ERK pathway and not different from targeting other nodes of the ERK pathway.

We note however that increased AKT phosphorylation (upon SHOC2 and/or ERK pathway inhibition in general) is clearly not sufficient to overcome the strong synergy observed upon combined SHOC2 KO and MEKi treatment, that is observed even in cell lines with co-occurring PIK3CA mutations such as HCT116 or H460 cells.

Minor:

I have not been able to locate where Figure S2C is cited in the manuscript.

This has now been corrected

Reviewer #3 (Remarks to the Author):

In this manuscript, Jones et al. discover a novel role for SHOC2 in promoting KRAS-driven anchorage independent growth, and synergy between SHOC2 depletion and MEK inhibition in KRAS and EGFR mutant lung cancer cell lines. First, they demonstrate impairment of lung tumor formation in Kras;p53 and Kras genetically engineered mouse models following SHOC2 deletion, and demonstrate that systemic SHOC2 ablation in adult mice is well tolerated over an 8 week period. Next, they uncover a specific impact of SHOC2 knockout on 3D cell growth and tumor xenograft formation or metastatic lung colonization in several KRAS mutant cell lines. More impressively they uncover potent synergy of SHOC2 deletion with MEK inhibition in multiple KRAS and EGFR mutant cell lines. Mechanistically they observe that SHOC2 loss prevents rebound pBRAF, pMEK, and pERK reactivation, and that this relates to disruption of MEKi induced RAF dimer formation. Finally, they demonstrate that combined suppression of SHOC2 and MEK inhibition induces apoptosis involving BIM, with impressive synergy in vivo in an A427 xenograft model.

Overall this is a very nice study that uncovers a novel target that synergizes with MEKi. I only have a few concerns that should be addressed prior to publication.

1. My major concern is the claim that SHOC2 deletion is well tolerated systemically, and that this therapeutic window is highlighted throughout the discussion. The authors are proposing MEK inhibitor combination studies, and no evidence is provided that SHOC2 deletion + MEK inhibition is tolerated systemically. Have the authors tried treating their SHOC2 KO mice with MEK inhibitors to see whether or not there is significantly increased toxicity?

We present data below from SHOC2 KO mice treated with the MEKi Selumetinib at 25mg/kg 5-days a week for 1 month to show that in this preliminary experiment we have not observed any toxicity as determined by loss of body weight or general body condition of the animals within this treatment window. This is consistent with the proposed potential for combined SHOC2 and MEK systemic therapies to provide an improved therapeutic window, but clearly still preliminary. Further studies using combined systemic SHOC2 and MEK inhibition will be the focus of future studies in the context of established RAS-driven tumour lesions and we would thus like to keep this preliminary data for a future follow up manuscript.

2. Along these same lines the MEF experiment in Figure 4F does show that SHOC2 deletion impairs P-ERK reactivation in the empty vector cells relative to KRAS G12V, except for the 30 min timepoint. Quantitation of P-ERK to total ERK signal would help to highlight the difference they are claiming in the absence or presence of KRAS G12V (though they do show that MEFs tolerate SHOC2 KO + MEKi relative to KRAS mutant cells in 3E).

The quantification of P-ERK on MEFs is now shown as Figure S6G. We also provide quantification for P-RSK in S6H that perhaps illustrates better a differential response between empty and KRAS-mutant cells. We also note that a differential effect between empty and KRAS MEFs is also observed at the level of P-MEK rebound and total BIM protein levels and shift.

As an alternative to treating the SHOC2 KO mice with MEK inhibitors, at a minimum it would be important to include additional data that human fibroblasts and/or AALE cells are less sensitive to the combination compared to lung cancer cells.

To address this point, we now include new Figure 3E where we use SHOC2 KO human NL20 bronchial non tumorigenic epithelial (generated by lentiviral CRISPR) and show that SHOC2 ablation does not have any significant effect on MEKi sensitization, consistent with data on RAS wt lung cancer cell lines. Furthermore, we also show that upon ectopic expression of KRAS^{G12V}, NL20 cells now become selectively sensitive to combined SHOC2 and MEK inhibition, thus broadly recapitulating the results on isogenic MEFs and further strengthening the selective sensitization to combined SHOC2 and MEK inhibition in the context of oncogenic RAS.

And regardless the discussion should have appropriate caveats that the therapeutic window of combined systemic SHOC2 and MEK inhibition is unclear.

We have now included in the discussion the underlined caveat

Although the toxicity of combined systemic SHOC2 and MEK inhibition in vivo remains to be addressed, our study suggests that uniquely among other pathway nodes for vertical inhibition, targeting the SHOC2 holophosphatase may overcome MEKi resistance in RAS-mutant ... with less toxicity.

3. The authors compare 2D and 3D effects of SHOC2 KO but only 2D effects of the MEKi combination. Does MEKi also impair 3D spheroid growth of KRAS mutant cell lines such as A549 and A427, which were resistant to SHOC2 KO?

We now include new Figure S3B to show that MEKi potently inhibit viability of 'SHOC2 3D resistant' A549 cells under non-adhered conditions (in fact more potently than under attached conditions) but SHOC2 ablation lowers the IC50 values to MEKis to a similar extent in both adhered and suspension culture conditions.

Minor Concerns

1. Figure 4G is referred to before Figure 4F in the text, figures should be called out in order.

This has been corrected.

2. Unclear why H522 data not shown? Having another KRAS WT example where pERK feedback is minimally suppressed by SHOC2 KO would strengthen the story, especially since the MEF data in Fig 4F is not entirely convincing.

H522 has been added to the manuscript as new Figure 4H (while the MEF data has now been moved to Figure S6F).

Reviewers' Comments:

Reviewer #1:

Remarks to the Author:

Summary: Despite the ubiquity of mutations in KRAS in non-small cell lung cancer (NSCLC), no effective anti-RAS therapies have transitioned into the clinic for NSCLC. Therefore, many groups have focused on the identification of pharmacologically accessible signaling pathways downstream of oncogenic RAS. In agreement with the requirement for MAPK pathway activation downstream of oncogenic RAS, small molecule inhibitors of each node have been approved or are under evaluation for therapeutic targeting in multiple cancers. Despite overwhelming evidence that targeting the RAF-MEK-ERK axis is efficacious in targeting hyperactivated MAPK pathway tumors, the responses are not durable due in large part to resistance. As described by Jones et al. the protein phosphatase SHOC2 is an instrumental regulator of RAS-mediated RAF kinase activation via dephosphorylation of a negative regulatory site within the kinases that prevents their dimerization. In the paper under review, Jones et al. seek to systematically interrogate the contribution of the SHOC2 phosphatase complex in KRAS-driven lung adenocarcinoma through the utilization of both a genetically engineered mouse model and human cancer cell lines to genetically ablate *Shoc2* or express an inactive mutant form of SHOC2, pharmacologically inhibit SHOC2 with a small molecule inhibitor, and test synergistic combinations for increasing the efficacy of targeting SHOC2. In simplistic but elegant experiments, the authors generate two novel strategies to conditionally inactivate SHOC2 function either through complete genetic deletion or expression of a minigene that encodes the SHOC2 D175N point mutant that were utilized to test the impact of SHOC2 dysfunction specifically in tumor tissue or in the entire animal. In addition, the authors corroborated their findings in human lung adenocarcinoma cell lines and established a potential model for how RAF regulation and subsequent MAPK pathway activation are altered in the context of SHOC2 inhibition. Together, these data support the notion that SHOC2 could be a therapeutic target in NSCLC. In new data provided in the revised manuscript and increased clarity provided in the rebuttal letter, the authors have successfully addressed the reviewers concerns.

Questions:

1. The authors provide convincing data that genetic loss of SHOC2 or expression of a SHOC2 mutant decrease KRAS-driven lung adenocarcinoma formation. However, the authors do not include the Kaplan Meier survival data for the SHOC2 knock in mice in the KRAS, p53 model. The authors should include these data to determine whether the knockin has a different phenotype in this model.

- In response to the reviewers comments, the authors have now explained in the rebuttal letter that due to technical issues with breeding the *Shoc2* knockin mice, they prioritized the LSL-KrasG12D;*Shoc2*KI mice instead of attempting to generate a large enough mouse colony to statistically compare the groups. In preliminary data from n=3 LSL-KrasG12D;*Shoc2*KI;Trp53flox/flox provided in the rebuttal letter, the authors suggest that loss of *Shoc2* phosphatase activity increasing the survival in the genetically engineered mouse model of KRAS-driven lung adenocarcinoma. Thus, the authors have successfully addressed this reviewer's concerns.

2. While the authors clearly show that knockdown of SHOC2 in a panel of human lung adenocarcinoma cell lines increases phosphorylation of BRAF P-S365, it is well appreciated that oncogenic KRAS utilizes the CRAF-BRAF heterodimer for MAPK signaling. The authors should repeat the experiments in Figure 2A and 2I to interrogate phosphorylation of CRAF P-S259 in order to determine whether SHOC2 loss also increases CRAF P-259 and contributes to the anti-tumorigenic properties (xenograft growth and 3D growth).

- In response to the reviewer's comment, the authors have now more thoroughly evaluated both CRAF and BRAF phosphorylation in the absence of SHOC2 in several KRAS mutation-positive human lung cancer cell lines. Specifically, in new data provided in Supplemental Figure 1A and Figure 4A and 4D the authors have now investigated the phosphorylation of CRAF at S259, which based on their hypothesis and previous work should be enriched in the absence of SHOC2. The

increase in P-S259 of CRAF in response to SHOC2 knockout can be easily visualized in Figure 4A and Figure 4D. However, it's hard to appreciate an increase in P-S259 in Supplemental Figure 1A. The authors should include a quantification of P-S259 of CRAF over total CRAF either below the blots or as a graph for these studies. This minor addition would successfully address this reviewer's concerns.

3. The authors suggest that the A427 cell is 'resistant' to SHOC2 genetic knockdown, while the cells when injected into tail vein form less tumors in the lung. The authors see differential phenotypes in the other cell lines that have phenotypes in 3D growth and xenograft experiments. While the authors attempt to address this with signaling experiments, each of the cell lines responds similarly in 2D and 3D in response to SHOC2 knockdown. The authors need to mechanistically address why these differential phenotypes occur. Is it based of a cell line mutational status or other signaling pathways? One of the striking differences in the 2D A427 line studies in the absence of SHOC2 is increased phosphorylation of AKT. Does this explain the differences?

- In response to the reviewer's comment, the authors took a more in-depth approach at interrogating both the genetics and the cell signaling in the SHOC2 knockout 'resistant' lines. As suggested by the reviewer, the authors found and provide new data in Supplementary Table 1 and Figure 2I that indicates the SHOC2 'resistant' lines harbor mutations in STK11/LKB1 and retain high levels of AKT phosphorylation. Moreover, in more mechanistic studies to address this differential response the authors provide new data in Figure 2J-L that demonstrate that the SHOC2 'sensitive lines' can be rendered insensitive by stable ectopic expression of myristolated AKT. Thus, the authors have successfully addressed this reviewer's concerns.

4. The authors indicated that "However, SHOC2 inhibition lowered IC50 values to MEKis to a similar extent in both adhered and suspension culture conditions (data not shown), and therefore viability assays were performed in subsequent studies under adhered conditions." These data are critical to the interpretation of the results in subsequent figures on the combination of SHOC2 knockdown and MEK1/2 inhibitor efficacy in 2D. These data not shown should be included in the manuscript.

- In response to the reviewer's comment, the authors provide the data not shown in Supplemental Figure 3B to demonstrate that either in 2D or 3D SHOC2 knockdown increases sensitivity to the MEK1/2 inhibitor selumetinib. Thus, the authors have successfully addressed this reviewer's concerns.

5. The authors do a well-designed genetic experiment in Figure 3I-J and Supp Figure 2B in which they knockout endogenous SHOC2 and overexpress phosphatase insensitive mutants of either ARAF, BRAF, or CRAF and test both MAPK signaling or sensitivity to MEK1/2 inhibitors. However, these experiments are done in the SHOC2 resistant cell line A427 in Figure 3I-J. What explains the enhanced sensitivity now when the signaling changes were the same in both of these lines?

- In response to the reviewer's comment, the authors explain that due to their nomenclature of SHOC2 resistant and sensitive cell lines, it was confusing to observe sensitivity to the MEK1/2 inhibitors in the absence of SHOC2. Presumably, the 'resistance' to SHOC2 ablation alone in these cells is due inefficient MAPK pathway inhibition. Thus, the authors have successfully addressed this reviewer's concerns. The authors should ensure this is adequately addressed in the text of the manuscript.

Reviewer #2:

Remarks to the Author:

All my concerns have been successfully addressed by the authors.

Reviewer #3:

Remarks to the Author:

The authors have done a nice job and have satisfactorily addressed my concerns.

REVIEWERS' COMMENTS:

Reviewer #1 (Remarks to the Author):

Summary: Despite the ubiquity of mutations in KRAS in non-small cell lung cancer (NSCLC), no effective anti-RAS therapies have transitioned into the clinic for NSCLC. Therefore, many groups have focused on the identification of pharmacologically accessible signaling pathways downstream of oncogenic RAS. In agreement with the requirement for MAPK pathway activation downstream of oncogenic RAS, small molecule inhibitors of each node have been approved or are under evaluation for therapeutic targeting in multiple cancers. Despite overwhelming evidence that targeting the RAF-MEK-ERK axis is efficacious in targeting hyperactivated MAPK pathway tumors, the responses are not durable due in large part to resistance. As described by Jones et al. the protein phosphatase SHOC2 is an instrumental regulator of RAS-mediated RAF kinase activation via dephosphorylation of a negative regulatory site within the kinases that prevents their dimerization. In the paper under review, Jones et al. seek

to systematically interrogate the contribution of the SHOC2 phosphatase complex in KRAS-driven lung adenocarcinoma through the utilization of both a genetically engineered mouse model and human cancer cell lines to genetically ablate Shoc2 or express an inactive mutant form of SHOC2, pharmacologically inhibit SHOC2 with a small molecule inhibitor, and test synergistic combinations for increasing the efficacy of targeting SHOC2. In simplistic but elegant experiments, the authors generate two novel strategies to conditionally inactive SHOC2 function either through complete genetic deletion or expression of a minigene that encodes the SHOC2 D175N point mutant that were utilized to test the impact of SHOC2 dysfunction specifically in tumor tissue or in the entire animal. In addition, the authors corroborated their findings in human lung adenocarcinoma cell lines and established a potential model for how RAF regulation and subsequent MAPK pathway activation are altered in the context

of SHOC2 inhibition. Together, these data support the notion that SHOC2 could be a therapeutic target in NSCLC. In new data provided in the revised manuscript and increased clarity provided in the rebuttal letter, **the authors have successfully addressed the reviewers concerns.**

Questions:

1. The authors provide convincing data that genetic loss of SHOC2 or expression of a SHOC2 mutant decrease KRAS-driven lung adenocarcinoma formation. However, the authors do not include the Kaplan Meier survival data for the SHOC2 knock in mice in the KRAS, p53 model. The authors should include these data to determine whether the knockin has a different phenotype in this model.

- In response to the reviewers comments, the authors have now explained in the rebuttal letter that due to technical issues with breeding the Shoc2 knockin mice, they prioritized the LSL-KrasG12D;Shoc2KI mice instead of attempting to generate a large enough mouse colony to statistically compare the groups. In preliminary data from n=3 LSL-KrasG12D;Shoc2KI;Trp53flox/flox provided in the rebuttal letter, the authors suggest that loss of Shoc2 phosphatase activity increasing the survival in the genetically engineered mouse model of KRAS-driven lung adenocarcinoma. **Thus, the authors have successfully addressed this reviewer's concerns.**

No action required

2. While the authors clearly show that knockdown of SHOC2 in a panel of human lung adenocarcinoma cell lines increases phosphorylation of BRAF P-S365, it is well appreciated that oncogenic KRAS utilizes the CRAF-BRAF heterodimer for MAPK signaling. The authors should repeat the experiments in Figure 2A and 2I to interrogate phosphorylation of CRAF P-S259 in order to determine whether SHOC2 loss also increases CRAF P-259 and contributes to the anti-tumorigenic properties (xenograft growth and 3D growth).

- In response to the reviewer's comment, the authors have now more thoroughly evaluated both CRAF and BRAF phosphorylation in the absence of SHOC2 in several KRAS mutation-positive human lung cancer cell lines. Specifically, in new data provided in Supplemental Figure 1A and Figure 4A and 4D the authors have now investigated the phosphorylation of CRAF at S259, which based on their hypothesis and previous work should be enriched in the absence of SHOC2. The increase in P-S259 of CRAF in response to SHOC2 knockout can be easily visualized in Figure 4A and Figure 4D. However, it's hard to appreciate an increase in P-S259 in Supplemental Figure 1A. **The authors should include a quantification of P-S259 of CRAF over total CRAF either below the blots or as a graph for these studies. This minor addition would successfully address this reviewer's concerns.**

This has now been included as Figure S1B. We would agree that changes in basal (Non-treated) S259 levels are hard to appreciate in Supplementary Fig1.A. What we do note and describe is that EGF stimulation induces BRAF/ CRAF S259 dephosphorylation, and that this RTK-induced dephosphorylation is blocked in SHOC2 KO cells. More generally when cells are treated with EGF, or as the reviewer points out, on MEKi treatment in Fig.4A-D, it is much easier to see the contribution of the SHOC2 phosphatase complex on S259 dephosphorylation and the text reflects this.

3. The authors suggest that the A427 cell is 'resistant' to SHOC2 genetic knockdown, while the cells when injected into tail vein form less tumors in the lung. The authors see differential phenotypes in the other cell lines that have phenotypes in 3D growth and xenograft experiments. While the authors attempt to address this with signaling experiments, each of the cell lines responds similarly in 2D and 3D in response to SHOC2 knockdown. The authors need to mechanistically address why these differential phenotypes occur. Is it based of a cell line mutational status or other signaling pathways? One of the striking differences in the 2D A427 line studies in the absence of SHOC2 is increased phosphorylation of AKT. Does this explain the differences?

- In response to the reviewer's comment, the authors took a more in-depth approach at interrogating both the genetics and the cell signaling in the SHOC2 knockout 'resistant' lines. As suggested by the reviewer, the authors found and provide new data in Supplementary Table 1 and Figure 2I that indicates the SHOC2 'resistant' lines harbor mutations in STK11/LKB1 and retain high levels of AKT phosphorylation. Moreover, in more mechanistic studies to address this differential response the authors provide new data in Figure 2J-L that demonstrate that the SHOC2 'sensitive lines' can be rendered insensitive by stable ectopic expression of myristolated AKT. **Thus, the authors have successfully addressed this reviewer's concerns.**

No action required

4. The authors indicated that "However, SHOC2 inhibition lowered IC50 values to MEKis to a similar extent in both adhered and suspension culture conditions (data not shown), and therefore viability assays were performed in subsequent studies under adhered conditions." These data are critical to the interpretation of the results in subsequent figures on the combination of SHOC2 knockdown and MEK1/2 inhibitor efficacy in 2D. These data not shown should be included in the manuscript.

- In response to the reviewer's comment, the authors provide the data not shown in Supplemental Figure 3B to demonstrate that either in 2D or 3D SHOC2 knockdown increases sensitivity to the MEK1/2 inhibitor selumetinib. **Thus, the authors have successfully addressed this reviewer's concerns.**

No action required

5. The authors do a well-designed genetic experiment in Figure 3I-J and Supp Figure 2B in which they knockout endogenous SHOC2 and overexpress phosphatase insensitive mutants of either ARAF, BRAF, or CRAF and test both MAPK signaling or sensitivity to MEK1/2 inhibitors. However, these experiments are done in the SHOC2 resistant cell line A427 in Figure 3I-J. What explains the enhanced sensitivity now when the signaling changes were the same in both of these lines?

- In response to the reviewer's comment, the authors explain that due to their nomenclature of SHOC2 resistant and sensitive cell lines, it was confusing to observe sensitivity to the MEK1/2 inhibitors in the absence of SHOC2. Presumably, the 'resistance' to SHOC2 ablation alone in these cells is due inefficient MAPK pathway inhibition. **Thus, the authors have successfully addressed this reviewer's concerns. The authors should ensure this is adequately addressed in the text of the manuscript.**

This has been addressed.

Reviewer #2 (Remarks to the Author):

All my concerns have been successfully addressed by the authors.

No action required

Reviewer #3 (Remarks to the Author):

The authors have done a nice job and have satisfactorily addressed my concerns.

No action required